# Pink-beam serial crystallography

A. Meents [1,2], M.O. Wiedorn [1,3], V. Srajer[4], R. Henning[4], I. Sarrou[1], J. Bergtholdt[1], M. Barthelmess[1], P.Y.A. Reinke[5], D. Dierksmeyer[1], A. Tolstikova[3], S. Schaible[2], M. Messerschmidt[6], C.M. Ogata[7], D.J. Kissick[7], M.H. Taft [5], D.J. Manstein[5], J. Lieske[2], D. Oberthuer[1], R.F. Fischetti [7] & H.N. Chapman [1,3,8]

Serial X-ray crystallography allows macromolecular structure determination at both X-ray free electron lasers (XFELs) and, more recently, synchrotron sources. The time resolution for serial synchrotron crystallography experiments has been limited to millisecond timescales with monochromatic beams. The polychromatic, "pink", beam provides a more than two orders of magnitude increased photon flux and hence allows accessing much shorter timescales in diffraction experiments at synchrotron sources. Here we report the structure determination of two different protein samples by merging pink-beam diffraction patterns from many crystals, each collected with a single 100 ps X-ray pulse exposure per crystal using a setup optimized for very low scattering background. In contrast to experiments with monochromatic radiation, data from only 50 crystals were required to obtain complete datasets. The high quality of the diffraction data highlights the potential of this method for studying irreversible reactions at sub-microsecond timescales using high-brightness X-ray facilities.

[1] Center for Free Electron Laser Science, DESY, Notkestrasse 85, 22607 Hamburg, Germany. [2] Deutsches Elektronen Synchrotron (DESY), Photon Science, Notkestrasse 85, 22607 Hamburg, Germany. [3] Department of Physics, University of Hamburg, Luruper Chaussee 149, 22761 Hamburg, Germany. [4] Center for Advanced Radiation Sources, The University of Chicago, 9700 South Cass Avenue, Argonne, IL 60439, USA. [5] Medizinische Hochschule Hannover (MHH), Institut für Biophysikalische Chemie, Carl-Neuberg-Str. 1, 30625 Hannover, Germany. [6] National Science Foundation BioXFEL Science and Technology Center, 700 Ellicott Street, Buffalo, NY 14203, USA. [7] Advanced Photon Source, Argonne National Laboratory, 9700 S. Cass Ave, Lemont, IL 60439, USA. [8] Centre for Ultrafast Imaging, Luruper Chaussee 149, 22761 Hamburg, Germany. Correspondence and requests for materials should be addressed to A.M. (email: alke.meents@desy.de)

Recent advances in highly brilliant X-ray sources, fast frame-rate detectors and novel sample delivery techniques have changed the way crystallographic data can be collected at both X-ray free electron lasers (XFELs) and third generation synchrotron sources[1,2]. Instead of rotating a large single crystal in the X-ray beam while acquiring a series of consecutive diffraction patterns, in serial crystallography (SX) datasets are collected by taking individual snapshots from hundreds to hundreds of thousands of microcrystals and then merging the diffraction data into a single and complete three-dimensional dataset.

Serial femtosecond crystallography (SFX) was developed at XFELs and more than a hundred structures have been determined using the approach, as revealed by the number of depositions to the protein databank[3]. Owing to the underlying "diffraction-before-destruction" principle, SFX allows applying orders of magnitude higher doses than in conventional macromolecular X-ray crystallography making it possible to obtain high-resolution structural information even from sub-micrometer sized crystals with exposure times in SFX in the 20–50 fs range[4–6].

At low-emittance synchrotron sources it has become possible to achieve extremely high photon flux densities, which enabled the expansion of the concept of SX to these facilities. In contrast to femtosecond sources, the dose for a given sample measured with synchrotron radiation is limited by radiation damage, which is most prevalent for microcrystals[7,8]. As in conventional crystallography, the tolerable dose can be extended by cryogenic cooling from about 150 kGy at room temperature[7,8] to more than 30 MGy[9,10]. With the serial approach, the reduced tolerable dose at room temperature can be compensated by measuring a larger number of crystals. To date, such experiments have been performed with synchrotron radiation monochromatized to a relatively narrow bandwidth of $\Delta E/E \approx 10^{-4}$. This is about 1/5th of the bandwidth of XFEL pulses ($\Delta E/E \approx 2 \times 10^{-3}$). According to simulations, the smaller bandwidth requires data collection from many more crystals than at XFELs as more snapshots are required to (randomly) sample the Bragg diffraction peaks with the narrower slices that are measured[11]. As a consequence, SX using monochromatic synchrotron radiation requires the measurement of typically tens of thousands of crystals with dimensions larger than 5 µm, and data collection typically takes several hours for a complete dataset[2,12].

So far SX experiments at synchrotrons have been performed with monochromatic radiation using millisecond exposure times and longer[2,12–18]. To track structural changes in irreversible enzymatic reactions such as proteolysis, phosphorylation, or decarboxylation proceeding on shorter timescales, shorter exposure times are required. A method that uses the full poly-chromatic spectrum of an undulator harmonic at a synchrotron radiation source is referred to as Laue diffraction[19,20]. By forgoing the monochromator and using only a mirror for high energy cut-off, a "pink beam" with a relative photon energy spread of $\Delta E/E \approx 5 \times 10^{-2}$ is obtained, with about 100 times more flux than the monochromatic beam. Likewise, increasing the bandwidth reduces the number of snapshots needed for SX, in turn reducing experiment time and sample consumption[11].

Many pioneering time-resolved experiments have been performed using this method of macromolecular Laue diffraction. These studied reversible photo-induced structural changes that could be repeatedly triggered by laser pulses, using large single crystals with time resolution down to the length of a single bunch, which is about 100 ps (full width at half maximum) at the advanced photon source (APS)[21–30]. Until now, such short polychromatic exposures have not been realized using micro-crystals, as required for SX measurement of irreversible reactions.

A well-known obstacle in Laue crystallography is the high-scattering background from the entire bandwidth of the incident beam compared with the Bragg peaks where only a small fraction of the spectrum contributes[31]. This typically results in small signal-to-noise-ratios $I/\sigma(I)$, thereby reducing the achievable resolution. This has been limiting the applicability of Laue crystallography especially in case of weakly scattering samples, such as microcrystals or crystals with large unit cells.

Although in the past, great effort was spent optimizing diffracted intensities $I$, such as by employing highly brilliant synchrotron sources, reduction of the corresponding noise level $\sigma(I)$ has been somewhat neglected. In particular, the reduction of the background level of diffraction patterns could significantly improve the measurements[32,33]. There are three main contributors to the background of a diffraction pattern: (1) detector noise; (2) diffuse or continuous scattering of X-rays by the crystal itself, either by coherent or incoherent scattering processes; and (3) scattering of X-rays by non-sample material. With the latest developments in X-ray detector technology providing single-photon sensitivity (Poisson-counting statistics), the first factor has been minimized. The second factor, diffuse or continuous non-Bragg diffraction and incoherent scattering is intrinsic to the sample.

The third factor, scattering of X-rays by non-sample material has two main contributors: the sample holder or transport medium, and air. We have recently developed a micro-patterned sample holder from single-crystalline silicon, which does not contribute to any background scattering, and allows for efficient removal of mother liquor[8,34]. Scattering of X-rays by air can be eliminated by performing diffraction experiments in vacuum[35]. As biological samples are prone to dehydration, this necessitates either cooling the samples to cryogenic temperatures or embed-ding them in a medium or enclosure[36,37]. Both approaches increase experiment complexity and limit studies to static struc-tures or introduce additional background sources. Our solution is to instead enclose the air path of the direct beam in capillary shields, both upstream and downstream of the sample and to flush the remaining not enclosed path of the direct beam with helium gas.

Using this approach providing very low background scattering levels, here we demonstrate room-temperature SX experiments with microcrystals using the pink X-ray beam from a synchrotron source with exposure times of 100 ps.

## Results

**Proteinase K and phycocyanin microcrystals**. Microcrystals from two different proteins were chosen as model systems to demonstrate our methodology: proteinase K with a molecular weight of 29.5 kDa and crystal sizes between 10 and 20 µm, and phycocyanin with a molecular weight of 37.4 kDa and crystal sizes between 30 and 40 µm. Proteinase K is an enzyme commonly used in molecular biology for nucleic acid preparations, in laundry detergents for its proteolytic activity, and is a potential system for future time-resolved mix-and-diffuse experiments. Phycocyanin is a cyanobacterial antennae protein, which is a part of the light-harvesting complex of the photosystem and is a potential candidate to study light-induced structural changes. The phycocyanin complex consists of two subunits, chains α and β (referred to as chains A and B) and appears to form disc-like hexamers of heterodimers $(\alpha_6\beta_6)$[38].

**Single X-ray pulse experiments**. Using our experimental setup and the serial data collection and processing strategy described in the "Methods" section, we collected high-quality diffraction patterns using single-pulse exposures of 100 ps duration per crystal with the pink beam of the BioCARS beamline at the

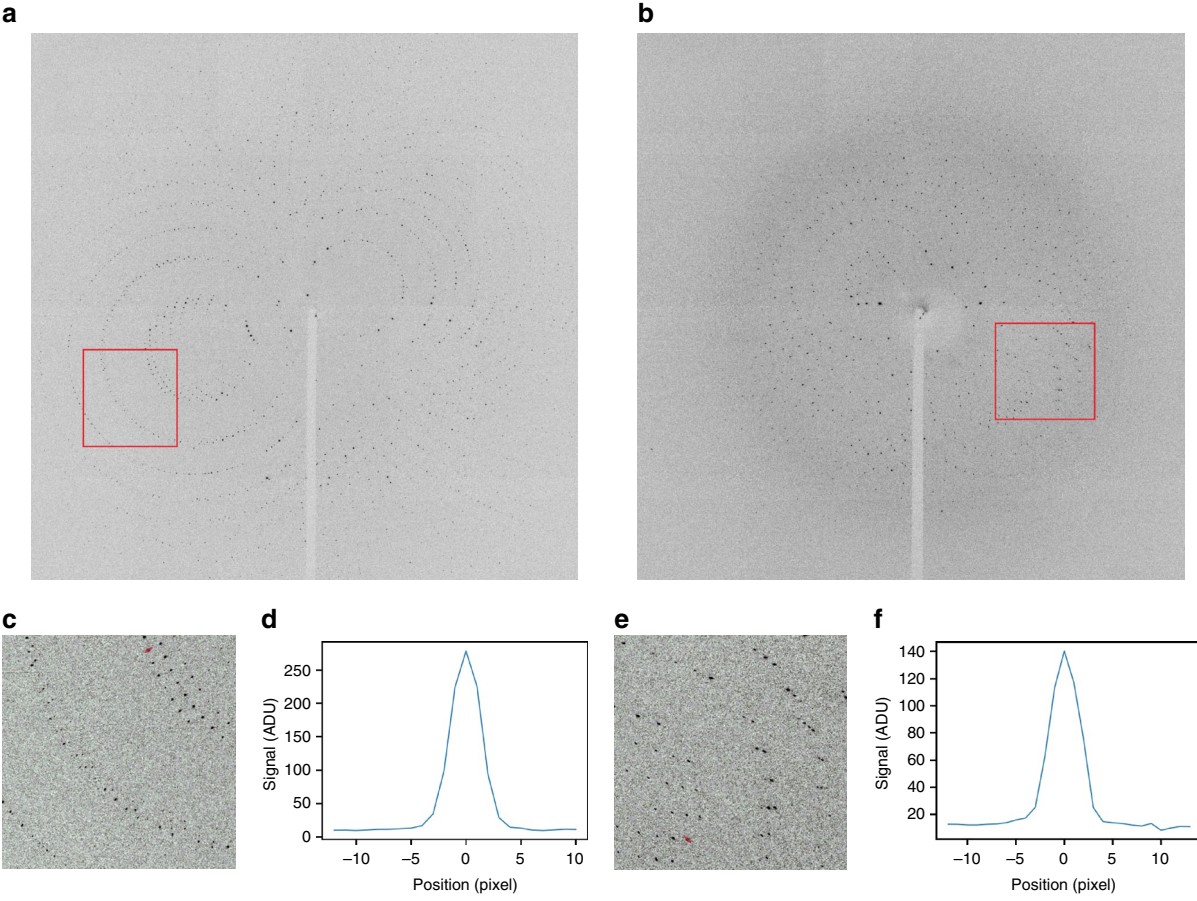

**Fig. 1** Exemplary single shot pink-beam diffraction images of **a** proteinase K and **b** phycocyanin microcrystals. Data were recorded at the BioCARS beamline using a Rayonix CCD detector. Crystal sizes were between 10 and 20 μm for proteinase K and 30 and 40 μm in case of phycocyanin. The diffraction images reveal the absence of a pronounced water ring that is typically observed in diffraction experiments from macromolecular crystals. **c**, **e** Insets of subfigures **a** and **b**, respectively, highlighting the shape of the Bragg spots. **d**, **f** Lineouts of the red traces in **c** and **e**, respectively, showing the shape of two exemplary Bragg spots from proteinase K and phycocyanin diffraction patterns

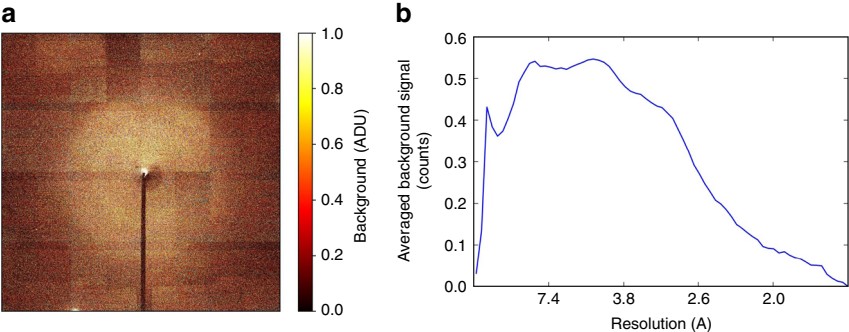

**Fig. 2** Quantitative analysis of the scattering background. Background scattering level from phycocyanin crystals mounted on a silicon chip (excluding diffraction intensities from Bragg peaks) highlighting the extremely low background level achievable with $3 \times 10^{10}$ photons/pulse (**a**). Radially averaged background level for the diffraction image as function of resolution (**b**)

APS. Examples of the resulting Laue diffraction patterns from proteinase K and phycocyanin crystals are shown in Fig. 1. The diffraction spots appear well separated and only slightly elliptical as seen in the insets and radial line profiles. The spots only have a small elongation in the radial direction, which implies a small degree of crystal mosaicity[39]. A large degree of crystal mosaicity, resulting in spots that are radially streaked, often leads to difficulties in data processing and is stated as a major limitation of Laue crystallography[40,41]. The diffraction patterns further reveal

the extremely low background levels achieved in our experiment. In particular, the absence of a so-called "water ring" at a resolution length of around 3 Å proves the efficient removal of excess liquid by using our silicon sample holder[34]. A quantitative analysis of the azimuthally averaged background scattering contribution is shown in Fig. 2, which further highlights the low background levels. For comparison, the noise level of the CCD detector used for the experiments is provided in Supplementary Fig. 1.

**Table 1 Data collection and refinement statistics for proteinase K and the different phycocyanin datasets**

|  | Proteinase K | Phyco_A1 | Phyco_A2 | Phyco_B | Phyco_C |
|---|---|---|---|---|---|
| Data collection | 5MJL | 5O7M | 5MJQ | 5MJM | 5MJP |
| Temperature | 293 K | 293 K | 293 K | 293 K | 293 K |
| X-ray pulse length | Single | Single | Single | Single | Multi-bunch |
| Dose per crystal (kGy) | 31 | 31 | 31 | 31 | 198 |
| Number of chips merged | 2 | 1 | 1 | 5 | 1 |
| Number of hits found | 1011 | 334 | 99 | 1072 | 601 |
| Indexed patterns | 140 | 91 | 47 | 364 | 115 |
| Number of lattices used in final merge | 59 | 45 | 40 | 205 | 52 |
| Space group | P43212 | R32:H | R32:H | R32:H | R32:H |
| Cell dimensions |  |  |  |  |  |
| $a, b, c$ (Å) | 68.3, 68.3, 108.3 | 187.8, 187.8, 60.7 | 187.8, 187.8, 60.7 | 187.8, 187.8, 60.7 | 187.8, 187.8, 60.7 |
| $\alpha, \beta, \gamma$ (°) | 90 | 90, 90, 120 | 90, 90, 120 | 90, 90, 120 | 90, 90, 120 |
| Resolution range (Å) (from data processing) | 100–2.21 (2.31–2.21) | 100–2.30 | 100–2.60 | 100–2.30 (2.4–2.30) | 100–2.12 (2.22–2.12) |
| Resolution range (Å) (used for completeness cut off) | 100–2.21 (2.31–2.21) | 100–2.46 (2.57–2.46) | 100–2.70 (2.82–2.70) | 100–2.30 (2.4–2.30) | 100–2.12 (2.22–2.12) |
| $R_{merge}$ | 0.054 | 0.040 | 0.041 | 0.057 | 0.061 |
| Mean $F/Sig(F)$[a] | 37.1 (25) | 47.6 (30) | 46.6 (26) | 47.2 (25) | 31.6 (23) |
| Completeness (%)[a] | 61.9 (26.1) | 60 (25.2) | 59.1 (26.9) | 67.1 (23) | 67 (26.6) |
| Wilson B | 0.02 | 8.86 | 13.59 | 12.00 | 15.21 |
| Refinement |  |  |  |  |  |
| Resolution (Å) | 44.1–2.21 (2.35–2.21) | 14.86–2.46 (2.55–2.46) | 14.90–2.70 (2.80–2.70) | 14.90–2.30 (2.39–2.30) | 14.9–2.11 (2.18–2.11) |
| No. reflections used | 8342 | 8904 | 6647 | 12,119 | 15,453 |
| No. reflections in $R_{free}$ | 793 | 892 | 666 | 1219 | 1510 |
| $R_{work}/R_{free}$ | 0.154 (0.172)/0.196 (0.220) | 0.142 (0.157)/0.196 (0.226) | 0.154 (0.202)/0.213 (0.182) | 0.139 (0.144)/0.181 (0.230) | 0.169 (0.170)/ 0.200 (0.182) |
| No. atoms |  |  |  |  |  |
| Protein | 2075 | 2498 | 2498 | 2498 | 2518 |
| Ligand/ion | 30 | 129 | 129 | 129 | 129 |
| Water | 322 | 67 | — | 127 | 180 |
| B-factors |  |  |  |  |  |
| Protein | 5.4 | 19.99 | 18.10 | 22.79 | 24.46 |
| Ligand/ion | 89.6 | 21.18 | 20.64 | 20.34 | 26.74 |
| Water | 10.8 | 23.45 | — | 26.72 | 34.27 |
| R.m.s. deviations |  |  |  |  |  |
| Bond lengths (Å) | 0.003 | 0.004 | 0.003 | 0.005 | 0.003 |
| Bond angles (°) | 0.65 | 0.54 | 0.54 | 0.60 | 0.52 |
| Clashscore | 1.21 | 3.26 | 2.88 | 2.50 | 3.04 |
| No. of TLS groups | — | 7 | 7 | 11 | 5 |

Phyco_A1, Phyco_A2, and Phyco_B datasets were measured with the APS single bunch in hybrid mode, Phyco_C with 3.68 μs exposures (single pulse plus eight septuplets in hybrid mode). Phyco_A1, Phyco_A2 and Phyco_C contain from a single chip only, whereas Phyco_B contains merged diffraction data from 5 chips (1, 6, 7, 8, 11)
[a]Values in parenthesis are for the outermost resolution shell

We solved the crystal structures of proteinase K and phycocyanin with pink-beam serial crystallography up to resolution of 2.21 and 2.12 Å, respectively. A detailed description of the data processing procedure and subsequent structure refinements are provided in the "Methods" section. Data collection and refinement statistics for proteinase K and for three different phycocyanin data collection scenarios are summarized in Table 1. We observe some variance for the ratio of "indexed patterns" and "number of hits found" between different chips. This is probably a result of different crystal densities on the chips leading to multiple hits in case of high crystal densities. Such multiple hits cannot be indexed with the current processing software.

A comparison of our proteinase K crystal structures derived from single-pulse Laue data with structural data obtained by conventional monochromatic single crystal diffraction experiments at room temperature (as deposited in the protein data bank with PDB codes: 2PRK and 4B5L)[42], reveals only small differences between the structures. Both datasets were collected from room-temperature samples. The structures are very similar to each other, as indicated by an overall rmsd value of 0.156 and 0.147 Å, respectively, for the main chain atoms of the superimposed molecules. The resolution of the single-pulse pink-beam data is not as high as for the single-wavelength data, as much larger crystals and longer exposure times were used in the monochromatic experiments. A comparison of the resulting exemplary electron density maps of proteinase K obtained by pink-beam serial crystallography with electron density maps originating from monochromatic data is shown in Fig. 3. Our pink-beam electron density maps show a high level of structural detail. The omit maps, obtained by removing residues 127–132 from the structure model, followed by simulated annealing refinement, further indicate that model-induced structural bias is absent (Supplementary Fig. 2). Nor do we observe any signs of specific radiation damage in the electron densities, such as in the example of a disulfide bridge of the proteinase K structure shown in Supplementary Fig. 3. A more detailed comparison of our proteinase K crystal structure quality parameters to those obtained from conventional single-crystal rotation photographs is provided in Supplementary Table 1.

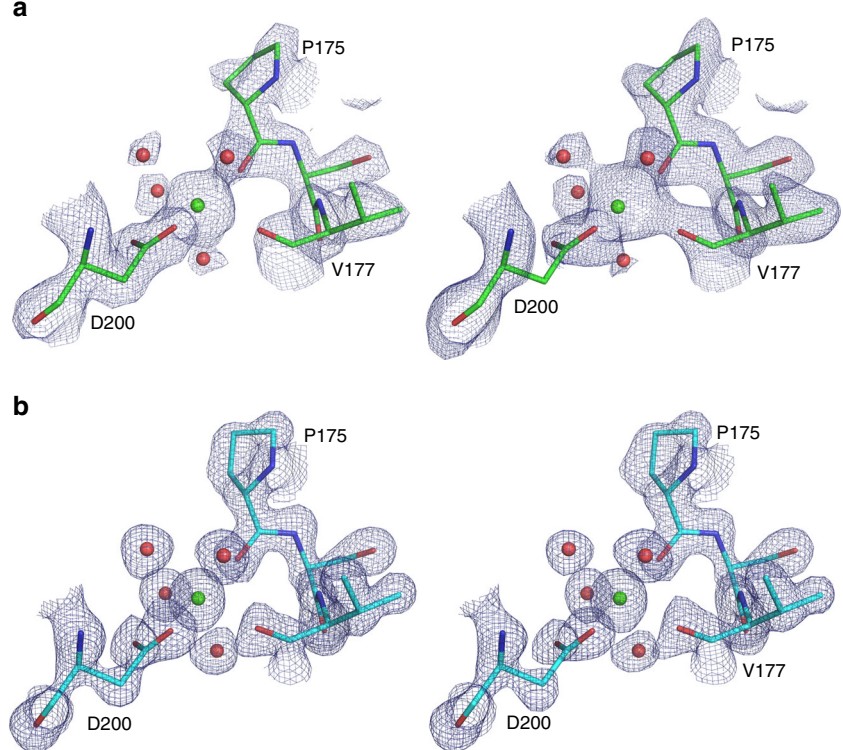

**Fig. 3** Comparison of electron density maps from proteinase K structures obtained from different diffraction methods showing the calcium binding site with the coordinating water molecules. **a** Structure obtained from our single shot pink-beam serial crystallography experiment. **b** Proteinase K structure from conventional single crystal rotation photographs (PDB ID: 2PRK). The blue grid represents $2mF_o - DF_c$ maps (left side) and $2mF_o - DF_c$ simulated annealing composite omit maps (right side) both at a contour level of $1.5\sigma$, respectively

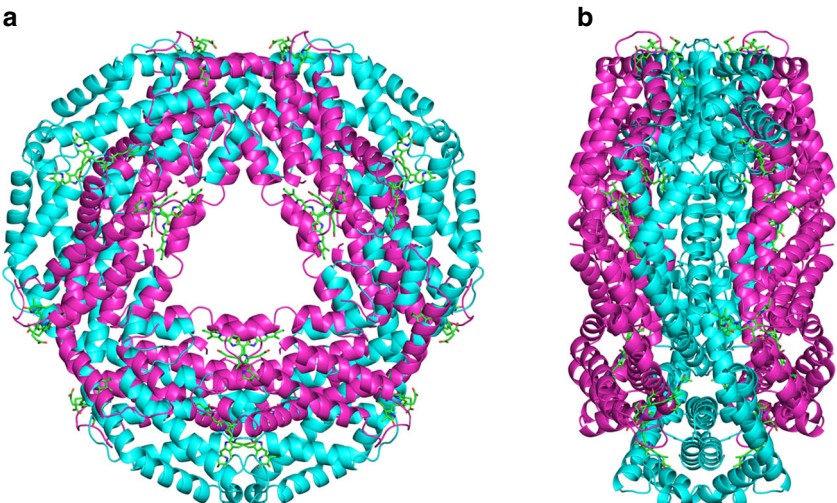

**Fig. 4** Overall structure of the biological assembly of phycocyanin: symmetry-related chains A in cyan, chains B in magenta displayed as cartoon. The phycocyanobilin molecules are shown in green. **a** View along the symmetry axis with **b** being a 90° rotation displaying the assembly from the side

**Data collection with multiple X-ray pulses**. Two distinct data collection scenarios were explored using phycocyanin to identify optimal conditions for pink-beam serial data collection and to examine potential radiation damage effects. In the first scenario, data were collected using single bunches in the APS hybrid mode with a pulse duration of 100 ps, resulting in a dose of 31 kGy. The second scenario used a longer exposure time of 3.68 µs (multibunch) resulting in a higher X-ray dose of 198 kGy. The corresponding filling pattern of the APS storage ring in hybrid mode is illustrated in Supplementary Fig. 4.

For the first scenario, datasets from five individual chips loaded with phycocyanin crystals were collected in total using single-pulse exposures. From these, three datasets were generated: Phyco_A1 and Phyco_A2 consist of data each collected from one single chip with data from 45 crystals and 40 crystals, respectively, in the final merge; Phyco_B includes data from all five chips with 205 crystals in the final merge including the data from Phyco_A1 and Phyco_A2. As a result of merging data from more chips and crystals into one single dataset, the achieved resolution increases from 2.46 Å for Phyco_A1 to 2.3 Å for Phyco_B. This clearly

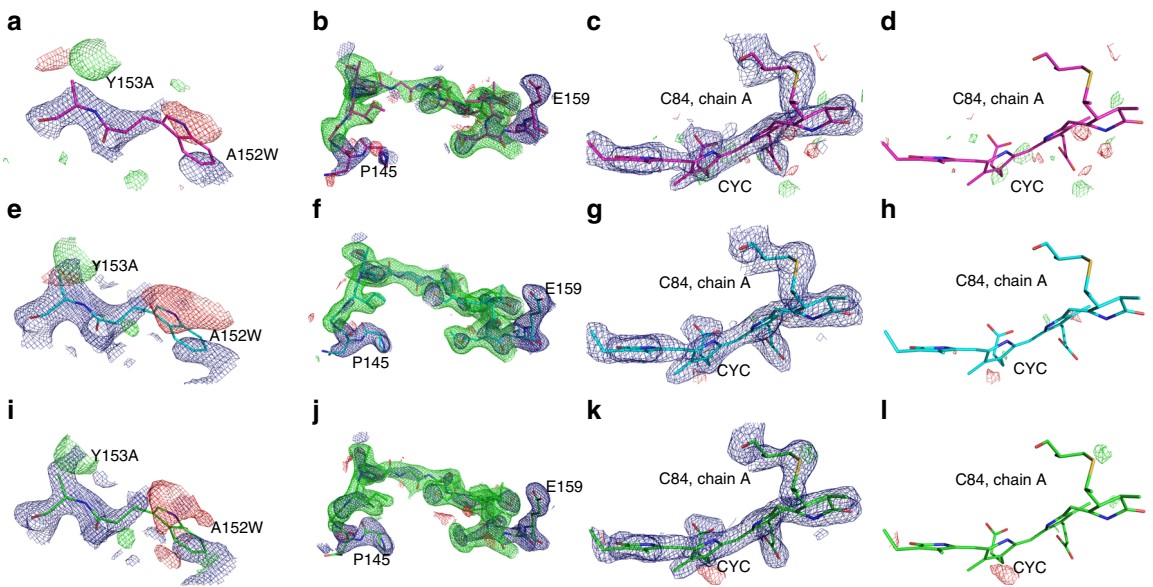

**Fig. 5** Difference electron density maps of different phycocyanin datasets. To assess the influence of phase bias introduced by the search model, we mutated residues 152 and 153 of chain A of Phyco A1 (single chip/single pulse, **a**, pink), Phyco B (five chips merged, single pulse, **e**, cyan), and Phyco C (single chip, multi-bunch exposure, **i**, green) and calculated simulated annealing maps (all blue maps are $2mF_o - DF_c$ maps at 1.5 sigma, all green and red maps are $mF_o - DF_c$ maps at 2.5 sigma). We furthermore deleted residues 147–157 of chain B for all three cases and calculated a simulated annealing omit map (**b**, **f**, **j**). Here we also show the omitted residues for clarity. In all three cases strong positive difference electron density is visible in the map in place of the omitted residues. To show that the mode of data collection does not lead to local radiation damage, we show the $2mF_o - DF_c$ map (**c**, **g**, **k**) and the $mF_o - DF_c$ map (**c**, **g**, **k**) and (**d**, **h**, **l**) for the bond between Cys84 of chain A and the phycocyanobilin-ligand for all three cases. No difference electron density is visible around the S–C-bond in any of the cases

highlights one of the basic concepts of serial crystallography where the achievable resolution as defined by $I/\sigma(I)$ can be improved up to the limit of the intrinsic crystal order by merging diffraction data from a large number of crystals.

In the second scenario of 3.68 µs exposures (multi-bunch), we collected a dataset from yet another chip referred to as Phyco_C. By merging data from 52 crystals mounted on a single chip, dataset Phyco_C contains the highest resolution diffraction data up to 2.1 Å. This can be explained by the factor of about 6.3 higher photon flux compared to measurements with the single 100 ps pulses. The Phyco_C multi-bunch dataset shows a slightly higher Wilson $B$-factor of 15.2 Å$^2$ compared to the 2.46 Å, 2.7 Å, and 2.3 Å datasets with Wilson $B$-factors of 8.86, 13.59, and 12 Å$^2$ for datasets Phyco_A1, Phyco_A2, and Phyco_B, respectively. This higher Wilson $B$-factor of the multi-bunch dataset could be explained either by thermal heating during the longer exposure or by global radiation damage effects due to the higher X-ray dose.

The overall structure of the biological assembly of phycocyanin is shown in Fig. 4. All phycocyanin electron density maps collected in the two scenarios appear to be of high quality. To assess the influence of potential phase bias introduced by the search model residues 152 and 153 of chain A of Phyco A1, Phyco B, and Phyco C were mutated and we calculated simulated annealing maps, which are shown in Fig. 5. The maps clearly reveal the mismatch between model and data introduced by the mutation in all three cases. In a second test, residues 147–157 of chain B were deleted, and simulated annealing omit maps were calculated. In all three cases strong positive difference electron density is visible at the position of the omitted residues, which supports the absence of phase bias in the starting model.

**Radiation damage**. To assess potential specific radiation damage effects we further calculated $2mF_o - DF_c$ and $mF_o - DF_c$ maps of the ligand-binding site of chain A (Cys84-CYC), which were carefully checked for residual electron densities. For all three

datasets, no significant residual electron densities could be found around the sulfur of Cys84 nor around the phycocyanobilin ligand (Fig. 5). In addition, we calculated $F_o - F_o$ maps, comparing the phycocyanin datasets obtained by serial Laue crystallography with high-resolution SFX (PDB code: 4ZIZ)[43] and synchtrotron datasets (PDB code: 1JBO)[44] to check for the presence of specific radiation damage in our data (Supplmentary Fig. 5; Supplementary Table 1). Again no residual difference electron density is present at a contour level of $3\sigma$ around any of the sulfur atoms in phycocyanin. Significant difference density is mainly visible around long flexible side chains and can be attributed to different average orientations of these side chains in the different datasets. The fact that we do not observe significant specific radiation, damage effects at doses of up to 198 kGy agrees well with previous results using monochromatic radiation, where no specific damage in insulin crystals is observed at doses up to 566 kGy[8].

**Comparison with other X-ray diffraction methods**. A comparison of exemplary electron densities for phycocyanin obtained with our method of serial pink-beam crystallography (datasets Phyco_A1, Phyco_B, and Phyco_C) with recent data obtained by SFX at the Linac Coherent Light Source (LCLS) is shown in Fig. 6. The SFX dataset (PDB code: 4ZIZ) consist of diffraction patterns from 6629 phycocyanin crystals with sizes of about 10 µm[43]. Interestingly, even though our pink-beam dataset is at lower resolution than the SFX data, the resolved structural features of the electron density appear similar. Simulated annealing composite omit maps for our datasets are also of high quality (Supplementary Fig. 6).

The biggest difference between the pink-beam datasets and the corresponding monochromatic and SFX datasets is the completeness, which is lower for the pink-beam data. Similar to resolution, the completeness of Laue data clearly improves when more crystals are merged (compare Phyco_B, 205 crystals and 67.1%

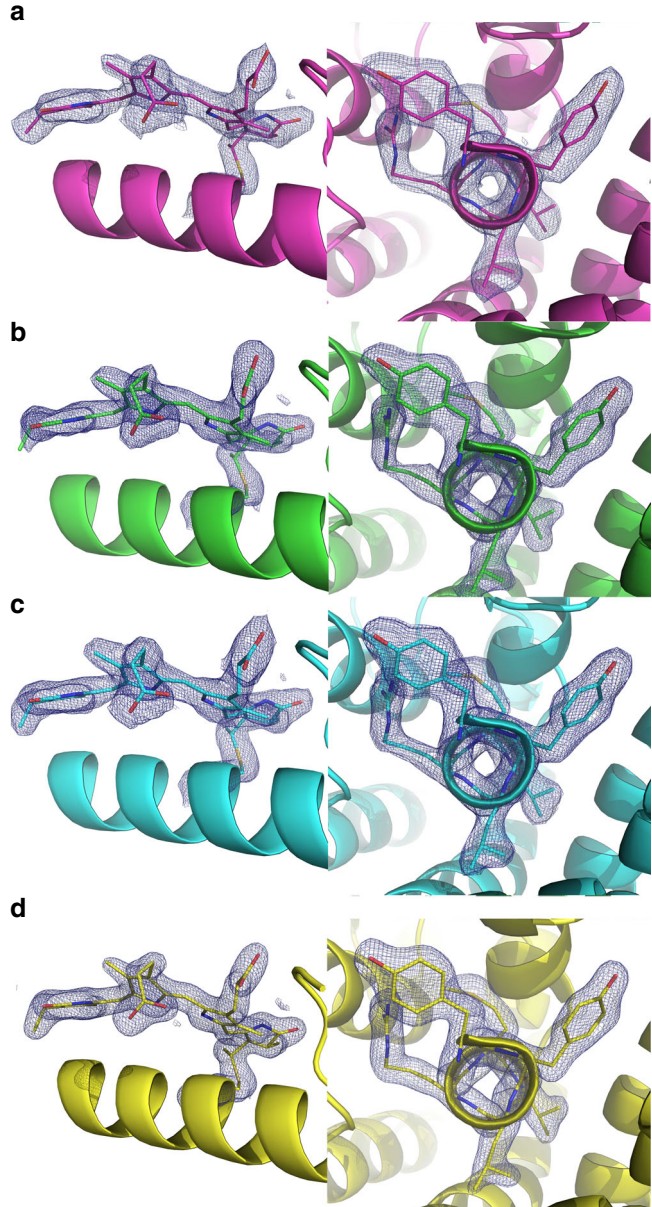

**Fig. 6** Comparison of exemplary electron density maps of phycocyanin ($2mF_o - DF_c$ at $1.5\sigma$ level) showing one of the phycocyanobilin molecules present in the structure (left) and a view along residues 90–95 of $\alpha$-helix 6 (right) obtained by pink-beam serial crystallography at a synchrotron (**a–c**) and with serial femtosecond crystallography (**d**). **a** The electron density of dataset Phyco_A1 obtained from the measurement of 45 diffraction images using the APS single bunch. **b** The electron density of dataset Phyco_B obtained from the measurement of 205 diffraction images using the APS single bunch. **c** The same regions obtained from 52 diffraction images using a longer exposure time of 3.68 μs per crystal. **d** Structure 4ZIZ from the PDB based on 6679 single shot diffraction images collected at LCLS

overall completeness, with Phyco_A2, 40 crystals and 59.1% overall completeness). Possible explanations for the reduced completeness at higher resolution are the more challenging data processing originating from angular overlap of diffraction spots, and a far more conservative resolution cut-off criterion of $3\sigma$ applied to the pink-beam data. This cut-off mainly affects the higher resolution reflections resulting in a lower completeness in these shells. The completeness as a function of resolution for all

datasets is shown in Supplementary Fig. 7. Lowering the cut-off criterion to $2\sigma$ led to a higher completeness but did not result in an improved data quality as judged by the refinement $R$-values. Hence, this cut-off was not used for further analysis.

Compared to monochromatic data, we obtain much lower Wilson B values in the analysis of the polychromatic data. This is most probably a systematic artifact of the Laue data reduction process and therefore these values should not be compared to Wilson B values from monochromatic data. The software Precognition preferentially excludes poorly measured weak intensities[20]. This leads to a flatter <I> curve of the Wilson plot at higher resolution values resulting in a lower $B$-value. For our proteinase K data, which was generally weaker than that of phycocyanin, this effect is even more pronounced (Supplementary Fig. 8).

## Discussion

By using polychromatic Laue diffraction in combination with a setup optimized for achieving extremely low background levels, we were able to obtain high-quality room-temperature SX diffraction datasets. The resulting electron density maps provide a high level of detail, similar to that achieved by using slightly smaller crystals with serial femtosecond crystallography at LCLS. Omit maps do not reveal any model bias induced by molecular replacement. The number of crystals for a complete X-ray dataset is significantly reduced to about 50–200 compared to the several thousands that are typically required for serial approaches using narrower bandwidth radiation. This makes this method well suited for cases where only a limited number of microcrystals are available. The significant reduction in the number of snapshots required also validates the approach of increasing bandwidth for XFEL-based serial crystallography[11].

By collecting data at room temperature, structural dynamics can be studied in a time-resolved fashion that otherwise be impeded by cooling[45]. A major challenge for the field of time-resolved X-ray crystallography remains the study of irreversible enzyme reactions, which can be initiated for example by diffusion of a substrate into crystals of the macromolecule[46]. Using SFX, the mechanism of Ribo-switching and the binding of an antibiotic to its target structure were recently revealed[47,48]. Such diffusion processes typically take place over timescales ranging from sub-milliseconds to seconds, and are best resolved using microcrystals because of much shorter diffusion times[46]. Crystal sizes were ranging from 10 to 20 μm in the case of proteinase K and 30–40 μm for phycocyanin, similar to the beam size of $20 \times 20$ μm$^2$ used in our experiment. Diffusion times of enzyme substrates into crystals of this size are ~10–20 ms. Our approach of pink-beam serial crystallography is well suited to perform such micro-diffusion experiments to study structural changes involved in enzyme reactions in a time-resolved fashion. By using the pink beam, such micro-diffusion reactions can be studied from measurements of significantly fewer crystals compared to monochromatic radiation, requiring much less beam time and crystalline material for a certain time step.

Given the low background observed in our experiments, it is clear that even smaller crystal volumes could be measured, at the expense of more snapshots required. It is interesting to consider the smallest crystal size that can reasonably be measured by single-pulse pink-beam crystallography. The signal-to-noise ratio of Bragg peaks in the diffraction patterns of phycocyanin recorded at a resolution of 2.3 Å with single 100 ps pulses was cut-off at $I/\sigma(I) = 3$. By including reflections at lower signal-to-noise ratios, for example down to 0.1, serial diffraction experiments with crystals about 1/30th of the volume, or about 3 μm crystal size, should be possible. Such an approach will require improved

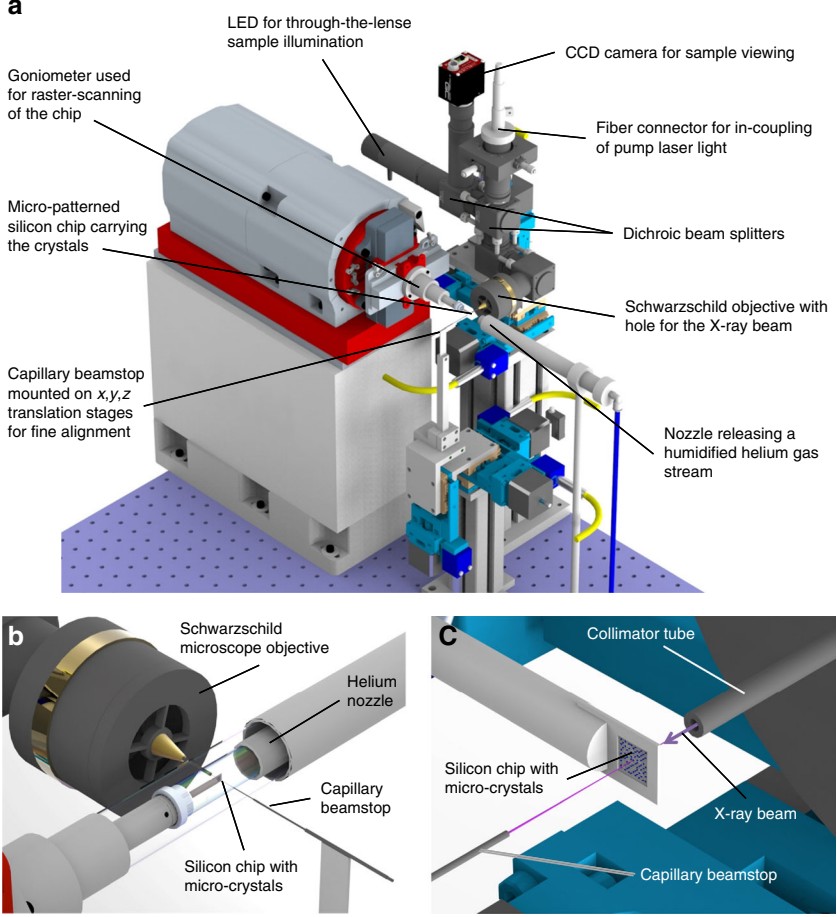

**Fig. 7 a** Experimental setup installed at the BioCARS instrument at the APS for fixed target serial crystallography using the pink beam. The micro-patterned silicon chip is raster-scanned through the X-ray beam using the goniometer installed at the beamline. An inline microscope using Schwarzschild optics is used for sample visualization. It provides the opportunity of through the lens illumination for both sample visualization and laser excitation of the sample. **b** Close-up showing the chip mounted on the beamline goniometer and being exposed to a stream of humidified helium gas preventing the crystals from drying out. **c** Schematic view of phycocyanin crystals mounted onto a micro-patterned silicon chip. The short path of the direct beam between the collimator tube and the capillary beamstop behind the sample is highlighted in pink. By flushing the remaining free beam path with helium, extremely low background scattering levels are achieved. The drawings were created using the software Solid Edge ST8, KeyShot 5.0, GIMP 2, and Microsoft PowerPoint

Laue data processing software. Crystals of this size are similar to their optical extinction depths, allowing optical lasers to more uniformly excite molecules and hence to achieve higher populations of the exited state during pump-probe experiments, compared to experiments using larger crystals. Pink-beam serial crystallography will extend the applicability of SX to 100 ps timescales, which are currently not accessible using monochromatic synchrotron radiation.

Serial crystallography breaks the nexus between dose and resolution. One of the defining aspects of the method is that the achievable resolution is not limited by crystal size or tolerable dose, but that the incident fluence and hence the X-ray dose can be reduced in favor of an increased number of snapshots to the degree that background allows. Our experiments demonstrate doses below tolerable levels with crystals small enough to support time-resolved experiments, for exposure times from 100 ps up to ~3.7 μs.

The maximum achievable data-acquisition rate was limited to about 2 Hz due to the step scanning approach used for our experiment. Using a fly-scan approach in combination with newest integrating fast frame-rate detectors, such as the Jungfrau or AGIPD detectors, should allow such experiments to be performed at much higher data-acquisition rates[33,49,50]. At the BioCARS beamline, such experiments will then be limited by the maximum frequency of X-ray pulses from the Jülich chopper, which currently is 987.4 Hz, or by the time structure of the synchrotron pulses. Assuming data collection at such frame rates and the corresponding hardware for fast sample delivery, a structure solution from the measurement of several tens of thousands of crystals will be possible in less than a second. This will allow both static and time-resolved structural investigations in a much more efficient way than currently done at any other synchrotron beamline.

Further improvements could be achieved using a better X-ray detection system. The background level in our experiments was dominated by thermal and readout noise of the detector system, which consists of a phosphor layer coupled to a CCD. Switching to a direct X-ray detection system, such as a silicon-based integrating detector developed for the XFEL sources, should allow the collection of almost noise-free data.

With the smaller source emittance anticipated by the lattice upgrade of the APS, for example, it will be possible to produce a focused spot of about 100 times higher intensity than used here, or about $10^{10}$ photons $\mu m^{-2} \mu s^{-1}$. This is similar to many SFX experiments using an attenuated beam, where crystal volumes of $1\,\mu m^3$ can easily be measured, despite the higher background (due to micrometer-thick liquid jets) compared to what can be achieved with our setup. There remains a question of tolerable

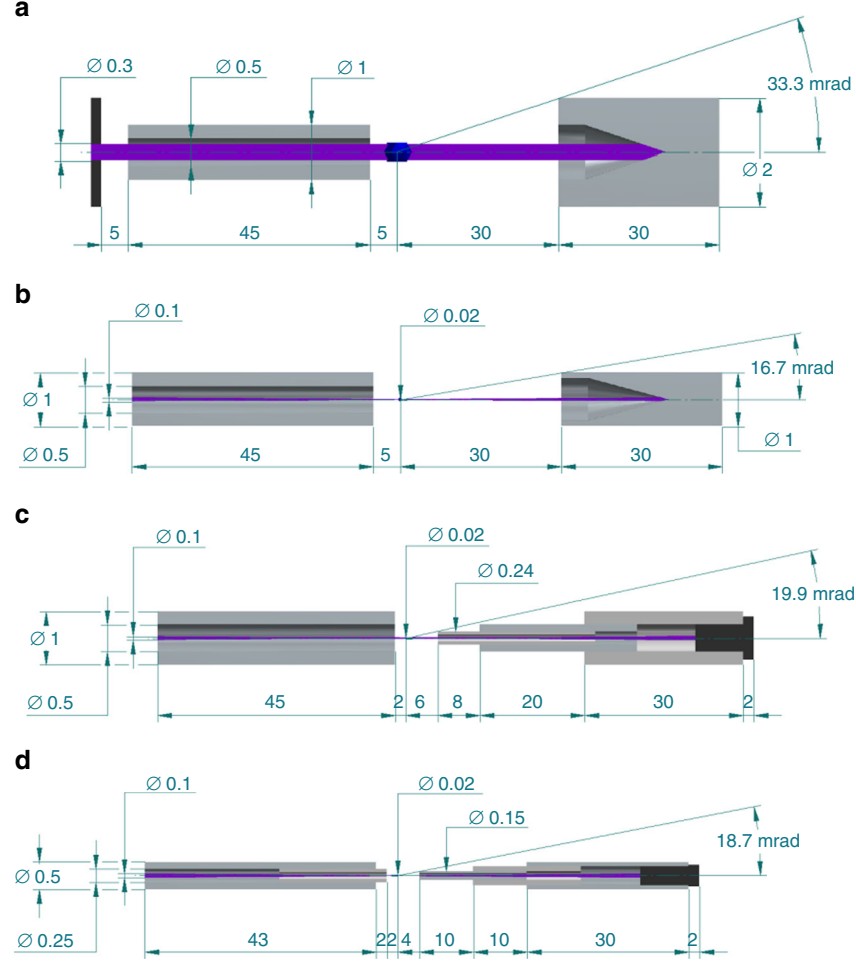

**Fig. 8** Evolution of collimator and beamstop designs. **a** Collimator and beamstop design typically used at protein crystallography beamlines at second generation synchrotron sources. A fraction of a relatively large X-ray beam is selected by using a 0.5 mm inner diameter collimator tube extending 5 mm close to the sample. After interaction with the sample, the direct beam is blocked with a 2 mm outer diameter metal cylinder with a blind hole placed about 30 mm behind the sample. The total path of the direct beam in air is about 35 mm. **b** Beamstop design currently used at most third generation synchrotron protein microfocus crystallography beamlines where much smaller samples can be investigated. Owing to the significantly smaller X-ray beam waist, the outer diameter of the beamstop can be one millimeter or less. Placed at a distance of again 30 mm low q-reflections can be now recorded up to a half opening angle of 16.7 mrad. The total path of the X-ray beam in air is again 35 mm. **c** Low background collimator and beamstop concept used for the present work. The path of the direct beam in air is reduced to 8 mm. X-ray photons scattered by air within the capillary are absorbed by the capillary walls. Compared to previous beamstop concepts **a**, **b** the path of the direct X-ray beam in air is shortened to 8 mm—reducing the background scattering level by a factor of about 4. At the end of the capillary the direct beam is blocked with a solid absorber. **d** Future beamstop concept: already 4 mm behind the sample the direct beam is fully enclosed by a 0.25 mm outer and 0.15 mm inner diameter metal capillary. The direct beam can be either blocked with a solid absorber or the tube can be extended through a hole in the detector and then blocked behind (favorable for XFEL applications). For data collection at cryogenic temperatures, collimator and post-sample beam-capillary would require heating, to prevent ice deposition caused by the cold-stream

dose under these conditions, but it is expected that an exposure time of 100 ps would outrun the diffusion of free radicals, calculated to recombine in nanoseconds[51,52], possibly giving a protection, similar to cryogenic cooling up to doses of about 30 MGy. Pink-beam serial crystallography can potentially provide high-throughput structure determination from microcrystals at room temperature, with an inherent time resolution of 100 ps.

## Methods

**Low-background experimental setup installed at BioCARS.** Diffraction experiments were performed at the BioCARS 14-ID beamline at the Advanced Photon Source (APS) in Argonne National Laboratory, USA[53]. Two in-line undulators at this beamline, with periods of 23 and 27 mm, provide high-flux polychromatic radiation at 12 keV.

The high photon flux, small beam size, and single-pulse isolation of the X-ray beam is achieved through a combination of choppers, a millisecond shutter, and a set of focusing X-ray mirrors. The first optical component is the heat load chopper that is used to reduce the heat load on all downstream components and to reduce

thermal drifts. The X-ray beam is focused by two Kirkpatrick–Baez (KB) mirror systems. The primary KB mirror system creates an initial focal spot size of $20 \times 70\ \mu m^2$ (vertically × horizontally) ~1.6 m upstream of the sample position. A secondary KB mirror system further reduces the beam size down to the final beam size of $20 \times 20\ \mu m^2$ as used for our experiment. A high speed Jülich chopper placed at the initial X-ray beam focus allows for single-pulse isolation. Sample exposure is controlled with the millisecond shutter.

To collect high-quality serial diffraction data from microcrystals using single polychromatic X-ray pulses, an experimental setup allowing to achieve very low background scattering levels was developed by us. It consists of four major components (Fig. 7a): (1) an inline sample-viewing microscope with through-the-lens sample illumination and the option of coupling laser light in for time-resolved experiments;[54] (2) the BioCARS goniometer for sample positioning; (3) a low background post-sample beam pipe replacing the conventional beamstop; and (4) a steel nozzle continuously releasing a gas stream of controlled humidity to prevent the crystals from drying out.

A major design goal of our setup was to eliminate air scattering, which is achieved by enclosing most of the path of the direct X-ray beam with thin-walled metal capillaries and by keeping the remaining unenclosed part of the beam path around the sample in a local helium atmosphere as proposed and demonstrated by

Roedig et al.[33] The number of X-ray photons scattered by air is proportional to the length of the beam path in air, which is about 35 mm in most crystallography experiments. We reduced this distance to 8 mm in our experiment, giving one quarter of the background scattering level onto the detector (Fig. 7b, c). This was achieved with a 0.5 mm inner diameter molybdenum tube placed in the axial bore hole of the inline-sample viewing microscope objective. The tube acts like a conventional collimating tube used at many crystallography endstations, except here it extended to a 2 mm distance before the sample. Furthermore, we replaced the conventional beamstop by three telescopic metal capillaries with inner and outer diameters that expand in sequence towards the detector (Fig. 8). With an outer diameter of 0.24 mm of the first capillary placed 6 mm behind the sample, this tube obstructs the low-angle information of the diffraction to a half angle of 20 mrad corresponding to a resolution of 54 Å, which is sufficiently low for most macromolecular diffraction experiments. The metal tubes completely shield the detector from air scatter generated by the enclosed X-ray beam.

By streaming helium gas across the remaining unenclosed direct beam path between the two capillaries at the sample position, the number of photons scattered by gas can be further reduced[33]. For 12 keV X-rays, this leads to a further reduction of the background scattering level by a factor of 26[55]. The continuous helium stream was generated by a 10 mm inner diameter nozzle directed towards the 8 mm unprotected sample region, in a direction roughly perpendicular to the X-ray beam. To prevent the crystals from drying out, the helium gas was maintained at a controlled humidity level by bubbling it through water as proposed by Roedig et al.[8] Combining the capillary shielding and humid helium schemes, the background scattering contribution was reduced by about a factor of 100 compared to a conventional macromolecular diffraction experiment. In principle, such a factor should reduce the noise, due to counting statistics of the background, by a factor of about $\sqrt{100} = 10$. Compared with the conventional setup, this reduction should allow data to be collected from a crystal of 1/10th of the volume to achieve the same signal-to-noise ratio for the same exposure and dose, or to reduce the dose by 10 times for the same sized crystal. This factor is in addition to the reduction in crystal size or dose obtained by adopting the serial approach, which can be orders of magnitude[56].

**Crystal growth**. Proteinase K from *Tritirachium album* (Carl Roth, Germany) was dissolved in CHC (Citric acid, HEPES, CHES) buffer (100 mM, pH 7) with 10 mM $CaCl_2$ added to yield a final protein concentration of 45 mg ml$^{-1}$ and filtered through a 0.2 μm filter (Sartorius Stedim, Germany) prior to crystallization. Crystals with sizes ranging from 10 to 20 μm in diameter were obtained at 20 °C by rapid mixing of one part of protein solution with one part of precipitant solution (1.6 M $MgSO_4$, 10 mM $CaCl_2$, 100 mM CHC buffer, pH 6.5) followed by incubation for 30 min.

Phycocyanin was isolated from the cyanobacteria *T. elongatus*. Whole cells were lysed by sonication in 25 mM MES (pH 6.4), 10 mM $CaCl_2$, 5 mM $MgCl_2$. After several washes at 12,000×*g*, the supernatant was further purified by ultracentrifugation at 100,000×*g* for 90 min. The new supernatant was concentrated and further purified using Amicon 100 kDa cut-off filters. PC was crystallized in 1.5 M ammonium sulfate solution, 25 mM MES pH 6.4. The protein crystals appear overnight in 1 mm inner diameter capillaries. The PC crystals size (30–40 μm) is correlated to the protein concentration, which varied between 10 and 15 mg ml$^{-1}$.

**Sample loading**. For the diffraction experiments, samples were loaded on micro-patterned silicon chips (Fig. 7c)[8,34]. Chips with different pore sizes ranging from 5 to 30 μm were used to achieve optimal coverage of the chips by matching the pore size to the crystal size. The spacing between the pores ranged from 10 to 50 μm. Owing to the use of single-crystalline silicon as substrate material, the efficient removal of excess liquid, and the absence of any window material, these sample holders do not contribute to any background scattering signal[34].

For sample loading, empty chips were mounted on the BioCARS goniometer and subsequently loaded with the sample by pipetting about 3 μl of a microcrystal suspension containing 2000–5000 microcrystals onto the chip[8]. Mother liquor was soaked-off through the pore by attaching a piece of filter paper. Dehydration of the crystals during loading and data collection was prevented by keeping the chips in a continuous helium stream at a controlled humidity level between 90 and 99.9%.

**Data collection**. Data were collected at the BioCARS beamline at the APS storage ring. During the measurements, APS was operated in "hybrid mode" with a fill pattern of a single electron bunch of 59 nC (16 mA), diametrically opposed to eight septuplets with a total charge of 317 nC (86 mA). For details about the fill mode see Supplementary Fig. 4. The single bunch was separated from the adjacent septuplets by 1.59 μs in both directions. The isolated single X-ray pulse (100 ps duration) in this mode delivers about $3 \times 10^{10}$ photons of about 12 keV to the sample (Supplementary Fig. 9). Most data collections were carried out using the single 100 ps pulse. With a measured beam size of $20 \times 20$ μm$^2$ (FWHM) at the sample position, this corresponds to a dose of 31 kGy, which is well below the long-exposure dose limit postulated for room-temperature X-ray diffraction experiments[8]. For comparison, phycocyanin data were additionally collected with 3.68 μs exposures, consisting of the 100 ps pulse plus the eight septuplets (multi-bunch), and $1.9 \times 10^{11}$ photons per crystal corresponding to a deposited dose of 198 kGy. Diffraction

images were recorded with a Rayonix MX340HS CCD detector in the standard 2 × 2 binning mode.

Data were collected in an automated fashion by raster scanning of the chip through the X-ray beam in a meander-like scan at a frequency of ~2 Hz. The scan grid was defined using the inline sample-viewing microscope. A grid size of 50 μm was chosen for the measurements. At each point in the scan the sample was exposed and the detector readout, irrespective of whether a crystal was in the exposed hole. The time needed to scan one chip, consisting of about 1000 individual measurements, was <10 min.

**Data processing**. Laue diffraction images were processed using the Precognition/Epinorm software (Renz Research Inc.) and the BioCARS Python script pyPrecognition developed for processing serial Laue crystallography data in a more automated manner. Data for each chip were initially processed separately. The script identifies "hits", i.e., images that contain diffraction patterns, based on a chosen set of criteria such us the number of detected spots above a certain count level compared to the background. The "hit" images are then indexed individually. Current Laue processing software such as Precognition, requires knowledge of the cell parameters and space group for indexing, so these were determined separately from room-temperature monochromatic data. Cell parameters were then refined for each Laue image and integration was done using analytical profile fitting, one of the integration options in Precognition[20]. Given the relatively small number of images per dataset, images were evaluated visually for correct indexing and for detecting multiple diffraction patterns. A sub-set of indexed images was thus selected for integration, containing only single-crystal diffraction patterns. The Laue diffraction images possess a relatively high spot density. Even for single-crystal diffraction patterns, spatial overlap of diffraction spots has to be resolved during integration. Presence of multiple diffraction patterns on a single image will result in inaccurate integration even if one of multiple patterns is correctly indexed and dominant. Data were integrated to sample-specific resolution limits as listed in Table 1. Data for each dataset were scaled and merged, retaining observations with $I/\sigma(I) > 3$. The final resolution was defined such that the highest resolution shell completeness is at least 25% (see data completeness in Table 1 and Supplementary Fig. 7). This cut-off value is chosen based on personal experience.

**Structure refinements**. MTZ-files for further processing were generated from HKL-files using f2mtz from the CCP4 software package[57]. Structure refinements were carried out using the PHENIX[58] program package.

**Proteinase K**. PDB entry 5AVJ[59] without solvent was used as an initial model, with all *B*-factors set to 20. Initially, the model was fit to experimental data via rigid body refinement, simulated annealing, coordinate refinement in real and reciprocal space as well as refinement of isotropic ADPs and occupancies using phenix.refine[60]. In the same step, water molecules were added automatically and the X-ray/stereochemistry and X-ray/ADP weights were optimized. This was followed by iterative rounds of manual model building in Coot[61] and refinement with phenix, omitting simulated annealing and rigid body refinement in later stages. At the final refinement cycle, hydrogen atoms were added to the model. Restraints and coordinates for the ligand molecules were gained by running eLBOW[62] providing the three-letter-code (NHE and EPE).

**Phycocyanin**. Coordinates from PDB entry 1JBO[44] were used as an initial model after modification (removal of alternate conformers and ordered solvent; re-setting of all isotropic B to 20). Initial refinement strategy using phenix.refine included rigid body refinement, simulated annealing, isotropic ADP and maximum likelihood reciprocal space refinement. This was followed by iterative rounds of manual re-building using coot[61] and refinement with phenix.refine[60]. In all cases, the refinement strategy included: isotropic ADP-, TLS-, real-space-, and maximum likelihood reciprocal space refinement. In all cases, Phyco_A2-ordered solvent was inserted in coot and refined using phenix.refine. ReadySet in phenix.refine was used to compute geometrical restraints of the phycocyanobilin-ligands. Additional restraints (1.82 Å with $\sigma$ of 0.1) were added manually in phenix.refine for the covalent bond between atom CAC of the phycocyanobilin-ligands and the sulfur atoms in CYSA84, CYSB82, and CYSB153 respectively. For all datasets the same set of reflections ($R_{\text{free}}$-flags) was excluded from refinement. Simulated annealing composite omit maps were calculated with PHENIX.

**Data availability**. Coordinates of the refined structural model and structure factors have been deposited to the protein data bank (pdb) with the accession codes: 5MJL (proteinase K), 5O7M, 5MJQ, 5MJM, and 5MJP (phycocyanin). Other data are available from the corresponding author upon reasonable request.

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

## Acknowledgements

This work was funded as part of the European Cluster of Advanced Laser Light Sources (EUCALL) project which has received funding from the European Union's Horizon 2020 research and innovation program under Grant Agreement No. 654220. This work was further supported by the European Research Council under the European Union Seventh Framework Program (FP/2007-2013)/ERC Grant Agreement No. 609920, "X-probe" funded by the European Union's 2020 Research and Innovation Program under the Marie Skłodowska-Curie Grant Agreement 637295, BMBF project 05K14CHA "Sync-FELMed", and the Virtual Institutes VI-403 and VI-419 of the Helmholtz Association. This research used resources of the Advanced Photon Source, a U.S. Department of Energy (DOE) Office of Science User Facility operated for the DOE Office of Science by Argonne National Laboratory under Contract No. DE-AC02-06CH11357. Use of Bio-CARS was also supported by the National Institute of General Medical Sciences of the National Institutes of Health under grant number R24GM111072. Time-resolved setup at Sector 14 was funded in part through a collaboration with Philip Anfinrud (NIH/NIDDK). The content is solely the responsibility of the authors and does not necessarily represent the official views of the National Institutes of Health.

## Author contributions

A.M., M.O.W., D.M., R.F.F., and H.N.C. designed the experiment. A.M., M.O.W., R.H., J.B., and D.D. designed and built the experimental setup. M.B. and S.S. prepared the fixed target sample holders. I.S., M.B., P.Y.A.R., S.S., C.M.O., D.J.K., M.H.T., and D.O. grew the crystals and prepared the samples for data collection. A.M., M.O.W., V.S., R.H., I.S., J.B., M.B., P.Y.A.R., D.D., A.T., S.S., M.M., D.O., and R.F.F. participated in the data collection. A.M., M.O.W., V.S., A.T., J.L., and D.O. analyzed the data. M.O.W., V.S., J.B., J.L., and D.O. prepared the figures for the manuscript. A.M., M.O.W., J.L., D.O., and H.N.C. wrote the manuscript.

## Additional information

**Competing interests:** A.M. is a CEO and shareholder of the DESY spin-off company Suna-Precision GmbH. Suna-Precision sells technical equipment for experiments with X-rays, including different microstructured silicon chips for serial crystallography experiments. The remaining authors declare no competing financial interests.

