## [Peer Review File · Nature Communications]

Reviewers' comments:

Reviewer #1 (Remarks to the Author):

The Meents paper introduces a new approach in serial synchrotron crystallography (SSX), whereby a polychromatic ('pink') beam is used to compensate both for the partiality issue inherent to monochromatic serial crystallography and for the relatively low flux of monochromatic synchrotron microbeams -- as compared to XFELs microbeams. The method holds the promise of enabling time-resolved SSX on the ns to μ s timescales, hitherto inaccessible to monochromatic SSX. Furthermore, and as the authors show, it would allow collecting SSX data from reduced amount of sample. Such novel work would deserve publication in Nature Communications.

Yet, it is the belief of this referee that in the present form, the manuscript is too drafty (writing) and approximate (results) to be accepted for publication. For example, the introduction and discussion could be reorganized and shortened (50 and 30% of the manuscript, respectively), leaving more space to present their results (20%). Additional calculations should be carried out to support the claims of the authors, notably those concerning the absence of radiation damage in their datasets or of model bias in their maps. Below, we highlight a number of specific issues. Hence, we recommend major revisions before the manuscript can be accepted for publication.

Line 38. "The X-ray exposures to single crystals" should read "The X-ray exposures of single crystals"

Line 42. "... must be offset by a higher exposure." should read "... must be compensated for by longer exposure (times)."

Line 48. "... under physiological conditions..." : it takes more than being at room temperature to claim that studies are performed in physiological conditions. Specifically, proteinase K and phycocyanin crystals were obtained using as precipitants 1.6 M MgSO₄ and 1.5 M NH₄SO₄, respectively. These crystallization conditions are far from physiological conditions.

Line 53: The Nango et al. 2016 (Science) paper describes results obtained on the ns to s timescales – that is not on the sub-ps timescale.

Line 53: "Here ..." is suggestive of results presented within this paper; use "There ..." or something else.

Line 38-56: The whole paragraph should be re-written to allow non-specialist readers to follow. Indeed, it is unclear what refers to SFX (serial crystallography at XFELs) or SSX (serial crystallography at synchrotrons). This referee advises to clearly state that the two methodologies exists, with SFX allowing data collection from sub-micron sized crystals and time-resolved studies on the ps-s timescale, whereas SSX is currently limited by crystal size (> 5-10 μ m) and may only enable time-resolved studies on the ns- μ s timescale by use of a polychromatic beam.

Line 63: "... for example diffusion of a substrate (such as ligand binding) ..." : these are two very different cases, and either should the authors focus on a single one of them or explain the differences between them. We note that it is unlikely that structural biologists will shift to complicated S(F/S)X experiments to simply solve the structure of a enzyme/ligand complex --- which is done routinely and efficiently using standard crystallography approaches. Thus, authors should rather focus on irreversible reactions involving substrate transformation into product(s) (i.e. structural enzymology).

Line 63: While we agree with the authors that the combination of the serial crystallography approach with rapid-mixing will become an important approach to study irreversible reactions in protein crystals, we respectfully disagree with the statement that "Serial crystallography" (in

combination with a rapid mixing injector) “has become the method of choice” to study ligand binding and associated conformational changes in crystalline proteins. To date, two serial papers have been published which made use of a rapid mixing device, and both described conformational changes occurring on the μ s timescale – that is, results which could have been obtained by use of standard crystallography and cryo-trapping. We believe it takes more than these two examples to reach the point where we can claim that S(F/S)X with mixing jets is a “method of choice”.

Line 67. “Such processes, taking place over timescales ranging from ms to s ...”. Which processes are the authors talking about? Do they refer to ligand or substrate binding? Regardless, they should note that these processes may occur on timescales shorter than the ms in diffusion limited enzymes. They should also better explain the rationale for using S(F/S)X and rapid mixing jets, so that non-specialist readers can follow; that is, they should state that trapping of intermediate states is complicated by the long diffusion times of solutes within protein crystals; that therefore, use of microcrystals is advised; that yet, diffraction from such crystals rapidly deteriorates due to radiation damage – both at RT and cryo-temperatures; and that hence, serial crystallography can save the day, because each crystal is exposed only once to the X-ray beam.

Line 73. We believe citations should reflect the variety of SSX approaches that have been used or proposed by different groups worldwide. To the contrary, the authors only cite 3 papers, two of which are from their group. We recommend correcting this, and citing at least Lyubimov et al. 2015 (Acta Cryst D), Coquelle et al. 2016 (Acta Cryst D), Huang et al. 2016 (Acta Cryst D) and Owen et al., 2017 (Acta Cryst D).

Line 80-83. “The smaller bandwidth requires data collection from many more crystals” (at the synchrotron) “than at XFELs”. The authors should state that they refer to simulation results, and not to experimental data. It could well be that due to photon energy jitter and to the use of detectors with lower dynamic range at XFELs, the effect due to the bandwidth is buffered out and SFX and SSX data end up being of the same overall quality.

Line 99. Although we could understand the sentence, “... such shots ...” does not refer to anything in the previous sentences. Please rephrase.

Line 100. “... resulting exposure times of many seconds”; that is only true for Laue experiments where microsecond exposures are used – which the authors introduce as a novelty in the next sentence. In the more common case (references 20-29 of the paper) where each crystal is exposed a 100 times (redundancy) over a 100 different orientations (completeness) for 100-1000 ps, our calculation is that the total exposure time is 1-10 μ s --- that is, far from the claimed exposure times of “many seconds”.

Line 112-118. We agree with the authors that high scattering background complicates processing of Laue crystallography data. However, the four main limitations of Laue crystallography are the requirements for (i) large crystals, (ii) small unit cells, (iii) high multiplicity and (iv) low mosaicity. Pink serial crystallography would address issues (i) and (iii); however, the need for relatively small and tightly-packed unit cells remains, as spot overlap due to large unit cell dimensions will inevitably complicate indexing and integration of Laue patterns, while high mosaicity (e.g. upon pumping by an actinic laser or mixing with a substrate) will result in a streaking of the Bragg spots that will prevent usage of the data.

Line 134-135. “... diffuse scattering ... bears the potential for overcoming the limitations of Bragg diffraction”. This is true at XFELs, where the molecule is static during the exposure, and could be true at synchrotrons, provided that the molecule is static during the exposure time. A pink beam would likely allow short enough exposures, but application of the diffractive imaging method described in Ayyer et al. 2016 (Nature) would likely not be feasible. Authors should either clarify this point or avoid sowing confusion by simply stating that diffraction imaging using protein crystals is at the present time only feasible in conjunction with SFX, at XFEL sources.

Line 146-149. "... to be placed in an enclosure." What are the authors referring to ? If vacuum chamber, then sure, it requires efforts, but this has obviously not limited success of experiments at LCLS and SACLA. If the authors are referring to the oil embedding of crystals and subsequent painting over a silicon nitride wafer as described in Hunter et al. 2017 (Sci Rep; ref. 37), then it would seem disproportionate to call this "considerable preparation and handling effort". As the authors later refer addition of "another window material", it is unclear if there is a confusion, because Hunter et al. only use a single Si₃N₄ wafer (no sandwiching as in Coquelle et al. 2015 (Acta Cryst D). These are furthermore virtually transparent to X-rays, so it wouldn't matter much.

Line 153. Again, even at room temperature, crystallization conditions are not physiological.

Line 180. Don't we expect the water ring at 3.8 Å instead ?

Line 186-193 and Table 1. Give doses for all datasets.

Line 207-208. "Our pink beam electron density maps provide ... no signs of radiation damage ...". The authors cannot just 'say' this. You claim it, prove it. Provide a Fo-Fo map, and state the Riso and/or CCiso.

Line 208-209. "The omit maps further reveal the absence of model induced bias". Same thing as above, you cannot claim this on the basis of a visual comparison between a 2FoFc map and a 2FoFc composite omit map, where you use the correct model as a phase source. If you want to use omit maps, then remove 10-20 residues and/or mutate residues in the structure, and report a (simulated annealing) FoFc omit map, which should show positive electron density around missing correct residues and negative electron density around the mutated ones.

Lines 195-212. In Supplementary Table 1, the authors should indicate integration and refinement statistics for the 4B5L structure, for which Rfree and completeness are available (contrarily to 2PRK) and which was refined with modern software and is therefore more comparable to theirs (refmac5 vs. phenix instead of pro-Isq vs. phenix). Also, the Wilson B values of their serial-pink proteinase K data differ in table 1 and supplementary table 1. This referee notes that both values are suspiciously low (0.02 (!!!) and 6.2 Å², respectively) – all the more for data collected at room-temperatures. Authors should at least give their explanation for these. They should also discuss what they believe is the significance of a 26.1 % complete highest resolution shell.

Lines 214-227 and 282-285. The authors collected three datasets from three chips, two with 100 ps and one with 3.8 μs exposures. Their data collection strategy allows to compare the effect of shooting 5 times more crystals or of using a 6.3 times higher photon flux, on the final quality of a pink serial dataset. They do show – without surprise -- that by adding more diffraction data, they get a better dataset. This referee suggests to go a step further and merge the phyco-A and phyco-B dataset to produce a more complete and therefore more meaningful (phyco-D) dataset. The latter should be used to compare the effect of exposure time in a rigorous fashion, i.e. by calculating FoFo maps between the phyco-C and phyco-D datasets, and between these and the available SFX data. Only then may the claim that the Phyco-C data shows not sign of radiation damage (line 282-285) be supported by data. This is all the more necessary when comparison of quality indicators in Table 1 suggests this dataset indeed suffered of radiation damage, with a higher Rmerge, a higher Wilson B and a reduced overall F/sigF. Use of higher quality phyco-D and SFX datasets will allow calculating FoFo maps with reduced noise levels. Authors should note that the Wilson B values reported in the text (lines 223-227) do not match those in Table 1 – which is again irksome as suggestive a hastily written manuscript.

Lines 236-237. Same remark as for lines 208-209. The presented data do not allow to make the claim there is no model bias. See above what should be done to indeed prove that there is no model bias in your maps. Note that in supplementary figure 4, it is indicated that the phyco-C data set

was collected with 40 μ s exposure...

Line 249-251. Have cut-off values intermediate between 2 and 3 been tried?

Line 258. "high levels of..." should read "high level of ..."

Line 259-260. See comments above regarding lines 236-237 and 208-209.

Line 268. "Crystal of this size match their optical extinction depths...". Again, the authors cannot just say that... First of all, at which wavelength? What is the protein concentration in crystals? What is the extinction coefficient of the protein at the wavelength of interest for the envisioned experiments? Assuming an epsilon of 100,000 at 620 nm (maximum absorption; Galzer et al. 1973 (JBC)), and a protein concentration of 25 mM in the crystals (as calculated from 4ZIZ), this referee calculates an optical extinction depth of 1.8 μ m – which would not match at all the size of the crystals.

Line 272-273. "... subnanosecond time scales, which are currently not accessible at synchrotron sources". 100 ps-time scale is accessible. Be specific.

Line 282-285. See comments above regarding lines 214-227 and 207-208.

Line 306. Reference 44 does not "show" that "monochromatic serial diffraction data can be processed ... signal to noise ratio of less than 0.1". Rather, it relates the data processing approach chosen by authors of that paper. But a clear-cut demonstration that inclusion of Bragg spots with signal to noise ratio of less than 0.1 meliorated the data is not present in the Gati et al. 2017 (PNAS) paper. This referee would suggest not including this reference, as informing readers that in previous work was used a $I/\sigma I$ cut-off of 0.1 is irrelevant to this report where was used a cut-off of 3.

Line 314-315. Authors suggest the use of direct X-ray detection systems in combination with the pink serial approach, but aren't such detector limited to $\sim 1 \mu$ s? How would they work with 100 ps exposure?

Line 319. "... produce a focused spot of ... 1×10^{10} photons/ μ m²". Per second?

Figure 1. Maybe would a temperature scale better show the Bragg spots in the exemplary diffraction patterns.

Supplementary information. Sample production and crystallization conditions should appear in the Method section.

Reviewer #2 (Remarks to the Author):

This article describes a powerful up-and-coming new method for pink beam crystallography that I expect will come into wide spread use over the next decade. It should be published with revisions.

Revision Notes:

Line 29: "consecutive rotation diffraction patterns" should be changed to "consecutive diffraction patterns as the crystal is rotated".

Line 32: "The quality of the data depends primarily on a high redundancy in the data, achieved by collecting a large number of patterns." In addition to redundancy (which is important), the quality of the depends most on the intrinsic order of the crystals. It is also affected by the level of non-isomorphism between crystals.

Line 45: "An attractive feature of serial X-ray crystallography is that by vastly reducing the required exposure per crystal it enables measurements at room temperature, even when using microcrystals." This is not correct for the XFEL. The attractive feature of serial crystallography at an XFEL is that though the 'diffraction before destruction' phenomenon a much larger dose may be applied to a small crystal as the diffraction pattern is obtained before most damage effects occur. For the synchrotron, the advantage is that a maximum dose may be applied to obtain a single diffraction pattern rather than the need to reduce the dose per single pattern to enable the collection of multiple diffraction patterns from a single crystal.

Line 53: "femtosecond-duration XFEL pulses" should be changed to "tens-of-femtosecond-duration XFEL pulses" The most common pulse duration used the LCLS is 40 fs (while sometimes shorter, 10 fs, pulses are used for diffraction.

Line 103: "With such large crystal volumes" should be "with crystals of this large volume,"

Line 105: "have not been demonstrated with microcrystals, as required for the measurement of irreversible reactions." Should be changed to "have not been recorded using microcrystals."

Line 109: What is a "Static serial crystallography experiment" Is this a 'fixed-target serial crystallography experiment', which is commonly defined as using a goniometer to position multiple micro-crystals, that are affixed to a rigid mount, in a serial fashion ?

109: "...which is around 100 ps" add "at the APS" or the word commonly or often: "which is often 100 ps". Some synchrotron sources offer use of smaller pulse durations for time-resolved studies.

124: "There are three main contributors to the background of a diffraction pattern: 1) readout and thermal noise of the detector" The state-of-the-art PAD detectors have minimal readout and thermal noise. However, there are still intrinsic errors related to the calibration of each pixel. I would suggest just saying "detector noise" rather than being specific to the cause.

128 "With the latest developments in X-ray detector technology providing both photon counting and integrating detectors with single-photon sensitivity (Poisson counting statistics) the first parameter can be regarded as overcome, given such a detector is available at the instrument." Even with these improvements, there are still significant errors from detector artifacts. This has been shown by offsetting PAD detector positions between the collection of multiple datasets from the same crystal, compared to collecting multiple datasets without offsetting a PAD detector. The errors in calibration of individual detector pixels is a major cause of error in diffraction measurements, even with PAD detectors. Therefore, I suggest changing the word "overcome" to "minimized" (However, even this may be an overly optimistic statement.)

208: "Our pink beam electron density maps provide a high level of structural detail and no signs of radiation damage are present." To prove there are no signs of radiation damage, please include a figure of electron density of a disulfide bond.

218: "This highlights one of the basic concepts of serial crystallography to improve the achievable resolution by merging diffraction data from a large number of crystals." This is only the case when the diffraction resolution of the crystals varies significantly, which often happens. The basic concept of serial crystallography is that the completeness and overall quality of the structure may be improved by merging diffraction data from a large number of crystals. – not necessarily the resolution.

270: after "... and hence to achieve higher populations of the excited state..." add ", during pump-probe experiments, " to clarify the purpose of the excitation.

272: "... to sub-nanosecond time scales," change to "to examine events over sub-nanosecond time

scales" or examine motions occurring over sub-nanosecond time scales.

275: add 'order of' to this sentence. "Furthermore, diffusion times of enzyme substrates into crystals with sizes below 30 μm are in the order of 10-20 ms" This is true for gasses and small substrates but could well be longer for larger substrates.

Figure 3: Could a difference map showing the density of the calcium be added to supplementary material. To avoid bias with making this difference map, the calcium should be removed from the molecular replacement and refinement model.

Figure 7: I have never encountered the setup described in A. If the X-ray beam was filling the tube as shown, I would expect significant spray/metal diffraction would result on the collected images. Maybe the setup in A should include a smaller slit up-stream of tube?

It should be noted that the distance at most synchrotrons is closer to 30 mm than 40 mm and this is for setups designed for data collection at cryogenic temperatures. It should also be mentioned that the setup described in figure 7 is not suitable for cryogenic data collection (due to the diameter of the cryo-streams currently used).

Table 1: Rmerge and MeanF/Sig(F) should be listed for the last shell. It would be useful to add CC1/2.

Reviewer #3 (Remarks to the Author):

Comments for hi5644

Overview and general points

The manuscript by A. Meents et al. describes a new serial data collection method using "pink beam of synchrotron radiation" targeting for microcrystals and time-resolve study. As protein crystals to be structure determined become smaller and smaller, radiation damage becomes a more serious problem, and data collection from one or a small number of crystals are becoming more difficult. Therefore, following the success of SFX of XFEL, research on serial data collection is currently reported as an important measurement method in the future even with synchrotron radiation. In this paper, they are trying to improve the efficiency of data collections by introducing a "pink beam" of an undulator harmonic with a wide energy bandwidth to serial crystallography. This is a certain success as a new trial of serial crystallography with synchrotron radiation. The manuscript can be evaluated as a proposal of a new measurement method.

Because it is difficult to understand the difference from the existing Laue method regrettably, what kind of technical improvement did serial crystallography corresponding to microcrystals realized? You should more emphasize the differences from existing technologies in terms of target crystal size and time resolution.

Although the manuscript suitable in general for publication, there are several questions and issues that need to be addressed by the authors before publication, as listed below.

Specific points

1. The advantage of using the pink beam is thought to be the improvement of the efficiency of data collection and the time resolution in the time resolved experiment with the increase of the photon flux per unit time, but it is easy to understand if there is a comparison of the experimental data with the monochrome beam.

Although it is not essential, I think that if you can discuss the exposure time per image and the

total experiment time required to realize similar data accuracy with monochromatic beam from comparative data, it would be a better paper

2. Around line 87-91 and from line 331 of "Methods"

On the description of the experimental setup of the "pink beam" corresponding to microcrystals, it is mainly described the view point of the reduction of the background scattering about the difference with the Laue setup on the same beamline (the BioCARS 14-ID at the APS) such as reference 30 (Sui, S. et al., 2016). I understand that it is important to reduce the background scattering, but has not improved the X-ray optical components and/or the optical setup corresponding to the microcrystals? For a better understanding it may be a method or a supplement, so more description should be given about the beamline and its optical system that made it possible to collect data from microcrystals

3. In line 243-245

As for the description of "Possible explanations for the reduced completeness at higher resolution are the more challenging data processing originating from an gular overlap of diffraction spots", I think that it is the influence of the streak at the diffraction spots by the pink beam, but what was the mosaic width of the crystal used for the measurement? The shape of the diffraction spot (the appearance of the streak) becomes easier to understand if there is a panel enlarging the diffraction spots in the diffraction image in Fig.1 with information of the crystal mosaics width. And I think that it is meaningful if there is a comment on the relation between the streak of diffraction spots and the crystal mosaic width when using a pink beam instead of white X-rays: more wide energy band width.

□ From line 434 "Data processing"

The energy spectrum of supplementary figure 2 is asymmetric tailing to the low energy side, but the asymmetry of the spectrum is taken into consideration in the integration process. I think that the asymmetry of the spectrum has an influence on the accuracy of the integrated intensity, but what would be the effect if it was considered at the integration process.

4. In line 403 "Data collection" and line 434 "Data processing"

According to Table 1, 500 to 600 hit images are excluded from 1000 measurement points per chip excluding Phyco_A, but only for Phyco_A why is 99 and much less?

On the other hand why the ratio of the index and the final merged image from the hit image is about 2-3 times the Phyco_A only? I guess it might have been that the crystal density on the chip was low only Phyco_A and the crystal overlap was small. For efficient data collection with "Pink Beam", it is important to improve the hit rate leading to the final data set, so I think it would be beneficial to have comments related to data processing

5. On Figure 6

Since the letters of the annotations in the schematic diagrams in Figs. 6 (b) & (c) are too small, you should devise measures such as enlarging the letters.

6. On Table1

As the last shell's completeness is low and the resolution limit is worrisome, so you should add "Mean F / Sig (F)" of the last shell to Table 1.

Since the notation of Rwork and Rfree in Table 1 is different between ProteinaseK and Phyco_X, please unify the notation.

Detailed response to referees letter

Reviewer Nr.	Comment	Reply to the comment
I.0	Yet, it is the belief of this referee that in the present form, the manuscript is too drafty (writing) and approximate (results) to be accepted for publication. For example, the introduction and discussion could be reorganized and shorten (50 and 30% of the manuscript, respectively), leaving more space to present their results (20%). Additional calculations should be carried out to support the claims of the authors, notably those concerning the absence of radiation damage in their datasets or of model bias in their maps. Below, we highlight a number of specific issues. Hence, we recommend major revisions before the manuscript can be accepted for publication.	As suggested by the reviewer and reported in more detail below we have now carried a much more detailed analysis of the diffraction data in particular with respect to model bias and potential radiation damage effects. We have extended the results section with these findings. We further provide a new figure 5 and added 3 more supplementary figures showing different difference electron density maps to support these claims.
I.1	Line 38. "The X-ray exposures to single crystals" should read "The X-ray exposures of single crystals"	Changes were made accordingly.
I.2	Line 42. "... must be offset by a higher exposure." should read "... must be compensated for by longer exposure (times)."	Changes were made accordingly.
I.3	Line 48. "... under physiological conditions..." : it takes more than being at room temperature to claim that studies are performed in physiological conditions. Specifically, proteinase K and phycocyanin crystals were obtained using as precipitants 1.6 M MgSO₄ and	Changes were made accordingly: "under physiological conditions" was changed to "room temperature"

	1.5 M NH ₄ SO ₄ , respectively. These crystallization conditions are far from physiological conditions.	
I.4	Line 53: The Nango et al. 2016 (Science) paper describes results obtained on the ns to s timescales – that is not on the sub-ps timescale.	The corresponding reference has been removed.
I.5	Line 53: “Here ...” is suggestive of results presented within this paper; use “There ...” or something else.	Changes were made accordingly.
I.6	Line 38-56: The whole paragraph should be re-written to allow non-specialist readers to follow. Indeed, it is unclear what refers to SFX (serial crystallography at XFELs) or SSX (serial crystallography at synchrotrons). This referee advises to clearly state that the two methodologies exists, with SFX allowing data collection from sub-micron sized crystals and time-resolved studies on the ps-s timescale, whereas SSX is currently limited by crystal size (> 5-10 μm) and may only enable time-resolved studies on the ns-μs timescale by use of a polychromatic beam.	The first two paragraphs have been restructured as suggested by the reviewer and serial crystallography at synchrotron sources has been introduced to make a clearer separation between the two techniques.
I.7	Line 63: “... for example diffusion of a substrate (such as ligand binding) ...” : these are two very different cases, and either should the authors focus on a single one of them or explain the differences between them. We note that it is unlikely that structural biologists will shift to complicated S(F/S)X experiments to simply solve the structure of a enzyme/ligand complex --- which is done routinely and efficiently using standard crystallography approaches. Thus, authors should	‘such as ligand binding’ has been removed from the sentence.

	rather focus on irreversible reactions involving substrate transformation into product(s) (i.e. structural enzymology).	
I.8	Line 63: While we agree with the authors that the combination of the serial crystallography approach with rapid-mixing will become an important approach to study irreversible reactions in protein crystals, we respectfully disagree with the statement that “Serial crystallography” (in combination with a rapid mixing injector) “has become the method of choice” to study ligand binding and associated conformational changes in crystalline proteins. To date, two serial papers have been published which made use of a rapid mixing device, and both described conformational changes occurring on the s timescale – that is, results which could have been obtained by use of standard crystallography and cryo-trapping. We believe it takes more than these two examples to reach the point where we can claim that S(F/S)X with mixing jets is a “method of choice “.	The corresponding sentences have been changed, as suggested, to: “A major challenge for the field of time resolved X-ray crystallography remains the study of irreversible enzyme reactions, which can be initiated for example by diffusion of a substrate into crystals of the macromolecule [44]. Using SFX, the mechanism of Riboswitching and the binding of an antibiotic to its target structure were recently revealed [45, 46]. Such diffusion processes typically take place over timescales ranging from sub-milliseconds to seconds, and are best resolved using microcrystals because of much shorter diffusion times [44].
I.9	Line 67. “Such processes, taking place over timescales ranging from ms to s ...”. Which processes are the authors talking about ? Do they refer to ligand or substrate binding ? Regardless, they should note that these processes may occur on timescales shorter than the ms in diffusion limited enzymes. They should also	We have rephrased the sentences and are now more specific: “ Such diffusion processes typically take place over timescales ranging from sub-milliseconds to seconds, and are best resolved using microcrystals because of much shorter diffusion times [44]. ” We further highlight now the potential of studying such

	better explain the rationale for using S(F/S)X and rapid mixing jets, so that non-specialist readers can follow; that is, they should state that trapping of intermediate states is complicated by the long diffusion times of solutes within protein crystals; that therefore, use of microcrystals is advised; that yet, diffraction from such crystals rapidly deteriorates due to radiation damage – both at RT and cryo-temperatures ; and that hence, serial crystallography can save the day, because each crystal is exposed only once to the X-ray beam.	processes using serial crystallography at synchrotron sources.
I.10	Line 73. We believe citations should reflect the variety of SSX approaches that have been used or proposed by different groups worldwide. To the contrary, the authors only cite 3 papers, two of which are from their group. We recommend correcting this, and citing at least Lyubimov et al. 2015 (Acta Cryst D), Coquelle et al. 2016 (Acta Cryst D), Huang et al. 2016 (Acta Cryst D) and Owen et al., 2017 (Acta Cryst D).	The suggested references have been added.
I.11	Line 80-83. “The smaller bandwidth requires data collection from many more crystals” (at the synchrotron) “than at XFELs”. The authors should state that they refer to simulation results, and not to experimental data. It could well be that due to photon energy jitter and to the use of detectors with lower dynamic range at XFELs, the effect due to the bandwidth is buffered out and SFX and SSX data end up being of the same overall quality.	We have changed the sentence to: “According to simulations, this reduction in bandwidth requires snapshots from more crystals to (randomly) sample the Bragg diffraction peaks with the narrower slices that are measured [9].”

I.12	Line 99. Although we could understand the sentence, "... such shots ..." does not refer to anything in the previous sentences. Please rephrase.	The corresponding passage has been rephrased to " Many pioneering time-resolved experiments have been performed using this method of macromolecular Laue diffraction. These studied reversible photoinduced structural changes that could be repeatedly triggered by laser pulses, using large single crystals with time resolution down to the length of a single bunch, which is about 100 ps at the Advanced Photon Source (APS) [19, 20, 21, 22, 23, 24, 25, 26, 27, 28]. "
I.13	Line 100. "... resulting exposure times of many seconds"; that is only be true for Laue experiments where microsecond exposures are used – which the authors introduce as a novelty in the next sentence. In the more common case (references 20-29 of the paper) where each crystal is exposed a 100 times (redundancy) over a 100 different orientations (completeness) for 100-1000 ps, our calculation is that the total exposure time is 1-10 μ s --- that is, far from the claimed exposure times of "many seconds".	See I.12
I.14	Line 112-118. We agree with the authors that high scattering background complicates processing of Laue crystallography data. However, the four main limitations of Laue crystallography are the requirements for (i) large crystals, (ii) small unit cells, (iii) high multiplicity and (iv) low mosaicity. Pink serial crystallography would address issues (i) and (iii); however, the need for relatively small and tightly-packed unit cells remains, as spot overlap due to large unit cell dimensions will	We agree with the reviewer that the requirements of Laue crystallography of using large crystals and datasets high multiplicity can be overcome with a serial approach. We further show that high quality diffraction data from Phycocyanin crystals can be obtained with our approach. With a unit cell volume of $2.14 \times 10^6 \text{ \AA}^3$ Phycocyanin cannot really be considered as a small unit cell system – even though there exist larger proteins. Crystal mosaicity is certainly a limitation

	inevitably complicate indexing and integration of Laue patterns, while high mosaicity (e.g. upon pumping by an actinic laser or mixing with a substrate) will result in a streaking of the Bragg spots that will prevent usage of the data.	of Laue crystallography as the reflections become streaky, spread out over many detector pixels, and it becomes difficult to obtain meaningful intensities at good S/N ratios. Again a low scattering background should be helpful here. During our experiments we rarely observed streaky reflections, which is most probably a consequence of using small crystals only. We fully agree, that when it comes to pumping or mixing experiments this will certainly affect the crystal mosaicity.
I.15	Line 134-135. "... diffuse scattering ... bears the potential for overcoming the limitations of Bragg diffraction". This is true at XFELs, where the molecule is static during the exposure, and could be true at synchrotrons, provided that the molecule is static during the exposure time. A pink beam would likely allow short enough exposures, but application of the diffractive imaging method described in Ayyer et al. 2016 (Nature) would likely not be feasible. Authors should either clarify this point or avoid sowing confusion by simply stating that diffraction imaging using protein crystals is at the present time only feasible in conjunction with SFX, at XFEL sources.	The corresponding reference has been deleted.
I.16	Line 146-149. "... to be placed in an enclosure." What are the authors referring to ? If vacuum chamber, then	Macromolecular diffraction experiments in an in-vacuum environment indeed require considerable

	sure, it requires efforts, but this has obviously not limited success of experiments at LCLS and SACLA. If the authors are referring to the oil embedding of crystals and subsequent painting over a silicon nitride wafer as described in Hunter et al. 2017 (Sci Rep; ref. 37), then it would seem disproportionate to call this “considerable preparation and handling effort”. As the authors later refer addition of “another window material”, it is unclear if there is a confusion, because Hunter et al. only use a single Si₃N₄ wafer (no sandwiching as in Coquelle et al. 2015 (Acta Cryst D). These are furthermore virtually transparent to X-rays, so it wouldn’t matter much.	preparation efforts. Hunter et al (2014) were using oil embedding to prevent the crystals from drying out. In Coquelle et al. (2015) crystals were placed between a Si₃N₄ sandwich, which also requires additional preparation effort, in particular if the sandwich structures have to be sealed for in-vacuum measurements (which was not the case here). In our case the sample is just pipetted on the chip and subsequently blotted. This takes typically less than a minute and doesn’t require transfer into an in-vacuum environment (additional handling effort). We have added a reference for a local enclosure for in-vacuum experiments of hydrated samples at an XFEL (Kimura et al., Nat. Comm. (2014)).
I.17	Line 153. Again, even at room temperature, crystallization conditions are not physiological.	We have changed ‘physiological conditions’ to ‘room temperature’.
I.18	Line 180. Don’t we expect the water ring at 3.8 Å instead ?	No, the O-O distance in pure water is typically around 2.8 Å at ambient conditions. For example see: Scott 2010, Water Journal. Depending on the buffer composition the observed values typically range from 2.8 Å – 3.3 Å. We have changed the phrasing and now refer to a “water ring at a resolution length around 3.0 Å.”
I.19	Line 186-193 and Table 1. Give doses for all datasets.	Doses were already given in the Methods part, but are now also provided in the Results section in the main text and also Table 1.
I.20	Line 207-208. “Our pink beam electron density maps provide ... no signs of radiation damage ...”. The authors cannot just ‘say’ this. You claim it, prove it.	It was not possible to calculate Fo-Fo maps for Proteinase K, as the structure the reviewer is referring to is not isomorphous, which is also expressed in

	Provide a Fo-Fo map, and state the Riso and/or CCiso.	different unit cell dimensions between the structures. In order to proof the absence of radiation damage, we have, as suggested by reviewer 2, calculated 2mFo-DFc maps of a disulfide bridge of Proteinase K, which is shown in supplementary figures 3. No signs of radiation damage can be observed in the structures. With a dose of 31 kGy per crystal we do not expect significant radiation damage, as this is well below the room temperature dose limits.
I.21	Line 208-209. "The omit maps further reveal the absence of model induced bias". Same thing as above, you cannot claim this on the basis of a visual comparison between a 2FoFc map and a 2FoFc composite omit map, where you use the correct model as a phase source. If you want to use omit maps, then remove 10-20 residues and/or mutate residues in the structure, and report a (simulated annealing) FoFc omit map, which should show positive electron density around missing correct residues and negative electron density around the mutated ones.	We have followed the procedure suggested by the reviewer. For this we have e.g. entirely removed residues 127 – 132 from the model of Proteinase K, refined the structure with simulated annealing, and subsequently calculated mFo-DFc maps. The results are shown in Supplementary figure 2. The positive electron density at the positions of the removed residues is clearly visible. We have included a reference to the new supplementary figure 2 in the manuscript.
I.22	Lines 195-212. In Supplementary Table 1, the authors should indicate integration and refinement statistics for the 4B5L structure, for which Rfree and completeness are available (contrarily to 2PRK) and which was refined with modern software and is therefore more comparable to theirs (refmac5 vs. phenix instead of pro-lsq vs. phenix).	The statistical information about the 4B5L structure has been added as requested by the reviewer. The 25% completeness cutoff of the data at the last resolution shell is arbitrary and based on the personal experience. We included this remark in the method section. We have further added the following paragraph to the

	Also, the Wilson B values of their serial-pink proteinase K data differ in table 1 and supplementary table 1. This referee notes that both values are suspiciously low (0.02 (!!!) and 6.2 Å², respectively) – all the more for data collected at room-temperatures. Authors should at least give their explanation for these. They should also discuss what they believe is the significance of a 26.1 % complete highest resolution shell.	Results section: “Compared to monochromatic data we obtain much lower Wilson B values in the analysis of the polychromatic data. This is most probably a systematic artifact of the Laue data reduction process and therefore these values should not be compared to Wilson B values from monochromatic data. The software Precognition preferentially excludes poorly measured weak intensities [23]. This leads to a flatter  curve of the Wilson plot at higher resolution values resulting in a lower B-value. For our proteinase K data, which was generally weaker than that of phycocyanin, this effect is even more pronounced (Supplementary Fig. 8).”
I.23a	Lines 214-227 and 282-285. The authors collected three datasets from three chips, two with 100 ps and one with 3.8 μs exposures. Their data collection strategy allows to compare the effect of shooting 5 times more crystals or of using a 6.3 times higher photon flux, on the final quality of a pink serial dataset. They do show – without surprise -- that by adding more diffraction data, they get a better dataset. This referee suggests to go a step further and merge the phyco-A and phyco-B dataset to produce a more complete and therefore more meaningful (phyco-D) dataset.	The Phyco_B dataset includes already the Phyco_A dataset, so this comparison is in principle done already. We have rephrased the corresponding paragraph, to make this more clearly to the readers.
I.23b	The latter should be used to compare the effect of exposure time in a rigorous fashion, i.e. by calculating FoFo maps between the phyco-C and phyco-D	We have calculated Fo-Fo maps between different Phycocyanin datasets. We have added a supplementary Table 2 showing the CC_iso and R_iso

	datasets, and between these and the available SFX data. Only then may the claim that the Phyco-C data shows not sign of radiation damage (line 282-285) be supported by data. This is all the more necessary when comparison of quality indicators in Table 1 suggests this dataset indeed suffered of radiation damage, with a higher Rmerge, a higher Wilson B and a reduced overall F/sigF. Use of higher quality phyco-D and SFX datasets will allow calculating FoFo maps with reduced noise levels. Authors should note that the Wilson B values reported in the text (lines 223-227) do not match those in Table 1 – which is again irksome as suggestive a hastily written manuscript.	values for all calculated maps. The statistical values for the Fo-Fo maps comparing the multibunch dataset (Phyco_C) with the high resolution SFX (4ZIZ) and synchrotron datasets (1JBO) are slightly worse than those where single pulse datasets (Phyco_A, Phyco_B) are compared to the SFX and synchrotron datasets. However upon inspection of the maps no significant difference density could be observed for both the Phyco_C vs 1JBO and Phyco_C vs 4ZIZ Fo-Fo-maps. The Wilson B values have been corrected.
I.24	Lines 236-237. Same remark as for lines 208-209. The presented data do not allow to make the claim there is no model bias. See above what should done to indeed prove that there is no model bias in your maps. Note that in supplementary figure 4, it is indicated that the phyco-C data set was collected with 40 μs exposure...	Please see our reply to comment I.22. Similar to Proteinase K we have calculated simulated annealing omit maps as suggested by the reviewer, which are shown in a new figure 5. No signs of modeal bias could be observed in the maps. The 40 μs exposure should be 3.68 μs and has been corrected.
I.25	Line 249-251. Have cut-off values intermediate between 2 and 3 been tried?	Values between 2 and 3 are generally possible but have not been tried as the effect between cutoffs at 2 and 3 sigma on the data quality were already relatively small.
I.26	Line 258. “high levels of...” should read “high level of ...”	Changes were made accordingly.
I.27	Line 259-260. See comments above regarding lines 236-237 and 208-209. Line 268. “Crystal of this size match their optical	The reviewer is right. We have changed the corresponding section

	extinction depths...”. Again, the authors cannot just say that... First of all, at which wavelength ? What is the protein concentration in crystals ? What is the extinction coefficient of the protein at the wavelength of interest for the envisioned experiments ? Assuming an epsilon of 100,000 at 620 nm (maximum absorption; Galzer et al. 1973 (JBC)), and a protein concentration of 25 mM in the crystals (as calculated from 4ZIZ), this referee calculates an optical extinction depth of 1.8 μm – which would not match at all the size of the crystals.	
I.28	Line 272-273. “... subnanosecond time scales, which are currently not accessible at synchrotron sources”. 100 ps-time scale is accessible. Be specific.	Changes have been made accordingly. The sentence now reads: “Pink beam serial crystallography will extend the applicability of SX to 100 ps time scales, which are currently not accessible using monochromatic synchrotron radiation.”
I.29	Line 282-285. See comments above regarding lines 214-227 and 207-208.	Please see reply to comment I20 and I23.
I.30	Line 306. Reference 44 does not “show” that “monochromatic serial diffraction data can be processed ... signal to noise ratio of less than 0.1”. Rather, it relates the data processing approach chosen by authors of that paper. But a clear-cut demonstration that inclusion of Bragg spots with signal to noise ratio of less than 0.1 meliorated the data is not present in the Gati et al. 2017 (PNAS) paper. This referee would suggest not including this reference, as informing readers that in previous work was used a $I/\sigma I$ cut-off of 0.1 is irrelevant to this report where was used a cut-	The corresponding reference has been deleted and the sentence has been rephrased accordingly.

	off of 3.	
I.31	Line 314-315. Authors suggest the use of direct X-ray detection systems in combination with the pink serial approach, but aren't such detector limited to ~1 μs ? How would they work with 100 ps exposure ?	The Jungfrau detector we are referring to is an integrating detector with single photon sensitivity for short exposure times (μs). This detector is similar to the CSPAD at LCLS designed for single shot diffraction experiments with short pulses. For details see e.g. reference 42. This is explained in more detail now in the manuscript.
I.32	Line 319. "... produce a focused spot of ... 1×10^{10} photons/ μm^2 ". Per second ?	The filling mode which was used for our experiments (hybrid mode) is not compatible with the new APS lattice after the upgrade. After the APS upgrade we expect a fluence of 1×10^{10} ph μm^{-2} μs^{-1} .
I.33	Figure 1. Maybe would a temperature scale better show the Bragg spots in the exemplary diffraction patterns.	We have tried different temperature scales but this didn't provide any additional information. Instead we decided to show now two magnified areas of diffraction spots on the detector and also to provide a line profile of one selected reflection per sample in figure 1.
I.34	Supplementary information. Sample production and crystallization conditions should appear in the Method section.	Changes were made accordingly
II.1	Line 29: "consecutive rotation diffraction patterns" should be changed to "consecutive diffraction patterns as the crystal is rotated".	The sentence has been changed to: "Instead of rotating a large single crystal in the X-ray beam while acquiring a series of consecutive diffraction patterns, in serial crystallography (SX) ...
II.2	Line 32: "The quality of the data depends primarily on a high redundancy in the data, achieved by collecting a large number of patterns." In addition to redundancy	In line with other reviewer comments, the introduction has been reordered and shortened. The passage the reviewer is referring to has been removed.

	(which is important), the quality of the depends most on the intrinsic order of the crystals. It is also affected by the level of non-isomorphism between crystals.	
II.3	Line 45: "An attractive feature of serial X-ray crystallography is that by vastly reducing the required exposure per crystal it enables measurements at room temperature, even when using microcrystals." This is not correct for the XFEL . The attractive feature of serial crystallography at an XFEL is that though the 'diffraction before destruction" phenomenon a much larger dose may be applied to a small crystal as the diffraction pattern is obtained before most damage effects occur. For the synchrotron, the advantage is that a maximum dose may be applied to obtain a single diffraction pattern rather than the need to reduce the dose per single pattern to enable the collection of multiple diffraction patterns from a single crystal.	We have restructured and changed the introduction section as already suggested by reviewer I (see reply to comment I.6). The differences between SFX and serial crystallography at synchrotrons are now described in more detail.
II.4	Line 53: "femtosecond-duration XFEL pulses" should be changed to "tens-of-femtosecond-duration XFEL pulses" The most common pulse duration used the LCLS is 40 fs (while sometimes shorter , 10 fs, pulses are used for diffraction.	We now explicitly mention that exposure times "in SFX are typically in 20 - 50 fs range."
II.5	Line 103: "With such large crystal volumes" should be "with crystals of this large volume,"	Changes were made accordingly.
II.6	Line 105: "have not been demonstrated with	We have changed the sentence to: "Until now, such

	microcrystals, as required for the measurement of irreversible reactions.” Should be changed to “have not been recorded using microcrystals.”	short polychromatic exposures have not been realized using microcrystals, as required for the measurement of irreversible reactions.”
II.7	Line 109: What is a “Static serial crystallography experiment” Is this a ‘fixed-target serial crystallography experiment’ , which is commonly defined as using a goniometer to position multiple micro-crystals, that are affixed to a rigid mount, in a serial fashion ?	‘Static’ in this case is referred to ‘static structures’. We have clarified this in the restructured introduction.
II.8	109: “...which is around 100 ps” add “at the APS” or the word commonly or often: “which is often 100 ps”. Some synchrotron sources offer use of smaller pulse durations for time-resolved studies.	Changes were made accordingly.
II.9	124: “There are three main contributors to the background of a diffraction pattern: 1) readout and thermal noise of the detector” The state-of-the-art PAD detectors have minimal readout and thermal noise. However, there are still intrinsic errors related to the calibration of each pixel. I would suggest just saying “detector noise” rather than being specific to the cause.	Changes were made accordingly. We now refer to ‘detector noise’ .
II.10	128 “With the latest developments in X-ray detector technology providing both photon counting and integrating detectors with single-photon sensitivity (Poisson counting statistics) the first parameter can be regarded as overcome, given such a detector is available at the instrument.” Even with these improvements, there are still significant errors from detector artifacts. This has been shown by offsetting	In this paragraph we are discussing the three main contributors to the background in a diffraction pattern. We agree that there are still issues with the calibration of individual detector pixels in particular for strong signals, but it has been shown in quite some cases that integrating PAD’s (in particular if they provide different gain stages such as the Jungfrau detector) provide zero background levels for short exposure times as in

	PAD detector positions between the collection of multiple datasets from the same crystal, compared to collecting multiple datasets without offsetting a PAD detector. The errors in calibration of individual detector pixels is a major cause of error in diffraction measurements, even with PAD detectors. Therefore, I suggest changing the word “overcome” to “minimized” (However, even this may be an overly optimistic statement.)	our case (see also reference 47).
II.11	208: “Our pink beam electron density maps provide a high level of structural detail and no signs of radiation damage are present.” To prove there are no signs of radiation damage, please include a figure of electron density of a disulfide bond.	The difference density map showing the disulfide bridge between residues C178 and C249 of Proteinase K has been calculated and is displayed in Supplementary figure 3. We have further calculated and inspected 2mFo-DFc difference maps of phycocyanin. In all structures no signs of radiation damage can be observed (see also reply to comment I.20).
II.12	218: “This highlights one of the basic concepts of serial crystallography to improve the achievable resolution by merging diffraction data from a large number of crystals.” This is only the case when the diffraction resolution of the crystals varies significantly, which often happens. The basic concept of serial crystallography is that the completeness and overall quality of the structure may be improved by merging diffraction data from a large number of crystals. – not necessarily the resolution.	We show in our manuscript that by merging more diffraction patterns collected at identical conditions and from the same crystal batch (dataset Phyco_A from 40 crystals compared to dataset Phyco_B from 205 crystals) the achievable resolution increases from 2.7 Å to 2.3 Å. This is also observed in more conventional SAD phasing experiments, where the resolution in terms $I/\sigma(I)$ typically increases with increasing data redundancy (e.g. by merging more patterns from more crystals, e.g. see).

		From a general statistical point of view, we can't change the measured intensities, but reduce the sigma by averaging more observables. To make this more clear to the readers, we have now define the resolution in terms $I/\sigma(I)$ in the corresponding sentence.
II.13	270: after "... and hence to achieve higher populations of the exited state..." add ", during pump-probe experiments, " to clarify the purpose of the excitation.	Changes were made accordingly.
II.14	272: "... to sub-nanosecond time scales," change to "to examine events over sub-nanosecond time scales" or examine motions occurring over sub-nanosecond time scales.	Changes were made accordingly, see also reply to comment I.28.
II.15	275: add 'order of' to this sentence. "Furthermore, diffusion times of enzyme substrates into crystals with sizes below 30 μm are in the order of 10-20 ms" This is true for gasses and small substrates but could well be longer for larger substrates.	We have changed the wording to "of this size are approximately 10 – 20 ms ." Referring to our crystal sizes and in line with the calculations presented in reference 44
II.16	Figure 3: Could a difference map showing the density of the calcium be added to supplementary material. To avoid bias with making this difference map, the calcium should be removed from the molecular replacement and refinement model.	An omit map showing the calcium site of Proteinase K is now displayed in supplementary figure 2, see also reply to comment I.20
II.17	Figure 7: I have never encountered the setup described in A. If the X-ray beam was filling the tube as shown, I would expect significant spray/metal diffraction would result on the collected images. Maybe the setup in A should include a smaller slit up-stream of tube?	A pinhole defining the beam diameter upstream of the collimator has been added to the illustration in figure 7 (now figure 8).

II.18	It should be noted that the distance at most synchrotrons is closer to 30 mm than 40 mm and this is for setups designed for data collection at cryogenic temperatures. It should also be mentioned that the setup described in figure 7 is not suitable for cryogenic data collection (due to the diameter of the cryo-streams currently used).	Changes were made accordingly and the distances have been changed 30 mm. The numbers have been also changed in the main text of the manuscript. We have further added a sentence to the figure caption, that the application of our novel beam stop concept to cryogenic temperatures would require heating of both the collimator and the post-sample beam-pipe in order to avoid ice-formation.
II.19	Table 1: Rmerge and MeanF/Sig(F) should be listed for the last shell. It would be useful to add CC1/2.	Rmerge and MeanF/Sig(F) for the outermost resolution shell have been added to Table 1. The function of calculating CC1/2 values for the outermost resolution shell is not available in Precognition.
III.0	Because it is difficult to understand the difference from the existing Laue method regrettably, what kind of technical improvement did serial crystallography corresponding to microcrystals realized? You should more emphasize the differences from existing technologies in terms of target crystal size and time resolution.	As requested by reviewer I (I.0 and I.6) we have restructured the introduction section and refer to conventional Laue crystallography in more detail now. In the following paragraph we emphasize our low background approach. Which is a pre-requisite to perform Laue-diffraction experiments with microcrystals.
III.1	1. The advantage of using the pink beam is thought to be the improvement of the efficiency of data collection and the time resolution in the time resolved experiment with the increase of the photon flux per unit time, but it is easy to understand if there is a comparison of the experimental data with the monochrome beam. Although it is not essential, I think that if you can discuss the exposure time per image and the total experiment time required to realize similar data	The general advantages of pink beam serial crystallography compare to a monochromatic approach are provided in the introduction. We now mention data collection times of two SX experiments carried out at a synchrotron using monochromatic radiation.

	accuracy with monochromatic beam from comparative data, it would be a better paper	
III.2	Around line 87-91 and from line 331 of “Methods” On the description of the experimental setup of the “pink beam” corresponding to microcrystals, it is mainly described the view point of the reduction of the background scattering about the difference with the Laue setup on the same beamline (the BioCARS 14-ID at the APS) such as reference 30 (Sui, S. et al., 2016). I understand that it is important to reduce the background scattering, but has not improved the X-ray optical components and/or the optical setup corresponding to the microcrystals? For a better understanding it may be a method or a supplement, so more description should be given about the beamline and its optical system that made it possible to collect data from microcrystals	The X-ray optical setup has not been changed between the experiments reported in Sui 2016 and ours. Sui et al. were just using a slightly larger beamsize of 35 x 35 μm^2 in combination with much larger crystal dimensions of $\sim 300 \mu\text{m}^2$. So the main difference is indeed the much lower background allowing us to collect high quality data from much smaller crystals. We have added a more detailed description of the X-ray optical beam path to the Methods section, which therefore had to be restructured.
III.3	In line 243-245 As for the description of “Possible explanations for the reduced completeness at higher resolution are the more challenging data processing originating from angular overlap of diffraction spots”, I think that it is the influence of the streak at the diffraction spots by the pink beam, but what was the mosaic width of the crystal used for the measurement? The shape of the diffraction spot (the appearance of the streak) becomes easier to understand if there is a panel enlarging the diffraction spots in the diffraction image in Fig.1 with	In our diffraction experiments from the different samples we could not observe significant streaking effects. This might be caused by using microcrystals, which often possess smaller mosaicities than larger crystals typically used for polychromatic diffraction experiments. To show this, two insets, showing magnified diffraction spots and exemplary line profiles in radial direction have been added to figure 1. This is now also mentioned in more detail in the manuscript.

	information of the crystal mosaics width. And I think that it is meaningful if there is a comment on the relation between the streak of diffraction spots and the crystal mosaic width when using a pink beam instead of white X-rays: more wide energy band width.	
III.4	From line 434 “Data processing” The energy spectrum of supplementary figure 2 is asymmetric tailing to the low energy side, but the asymmetry of the spectrum is taken into consideration in the integration process. I think that the asymmetry of the spectrum has an influence on the accuracy of the integrated intensity, but what would be the effect if it was considered at the integration process.	Yes, the asymmetry of the spectrum is taken into account during the integration process. As a consequence reflections originating from long wavelength tail of the spectrum are typically weak and as a consequence exhibit a lower signal-to-noise ratio. By measuring a larger number of crystals e.g. by means of serial crystallography reflections are typically measured multiple times with different wavelengths thus providing better S/N ratios.
III.5	In line 403 “Data collection” and line 434 “Data processing” According to Table 1, 500 to 600 hit images are excluded from 1000 measurement points per chip excluding Phyco_A, but only for Phyco_A why is 99 and much less? On the other hand why the ratio of the index and the final merged image from the hit image is about 2-3 times the Phyco_A only? I guess it might have been that the crystal density on the chip was low only Phyco_A and the crystal overlap was small. For efficient data collection with “Pink Beam”, it is important to improve the hit rate leading to the final data set, so I think it would be beneficial to have comments related to	The ratio between the indexed patterns to hits varies between 14% and 47%. For the chips with a low indexing rate we observed a significant fraction of multiple hits which can currently not be processed by the software. We have added the sentences: “We observe some variance for the ratio of ‘indexed patterns’ and ‘number of hits found’ between different chips. This is probably a result of different crystal densities on the chips leading to multiple hits in case of high crystal densities. Such multiple hits cannot be indexed with the current processing software. “

	data processing	
III.6	On Figure 6 Since the letters of the annotations in the schematic diagrams in Figs. 6 (b) & (c) are too small, you should devise measures such as enlarging the letters.	Changes were made accordingly.
III.7	On Table1 As the last shell's completeness is low and the resolution limit is worrisome, so you should add "Mean F / Sig (F)" of the last shell to Table 1. Since the notation of Rwork and Rfree in Table 1 is different between ProteinaseK and Phyco_X, please unify the notation.	Changes were made accordingly. The additional information has been added to the manuscript.

REVIEWERS' COMMENTS:

Reviewer #1 (Remarks to the Author):

We are pleased to recommend publication of the Meents et al. manuscript. The major claim of the paper, i.e. feasibility of polychromatic serial crystallography, is novel and will be of interest to the whole structural biology community. The work is convincing and we do not believe that more work is anymore needed to substantiate the authors claims. The authors have indeed addressed the issues we raised on the initial version of their manuscript. The paper now appears suited for publication, with three minor modifications.

- line 46. We believe citations should reflect chronological order. Feasibility of structure determination from sub-micron sized-crystals was demonstrated before Gati et al. 2017 PNAS. Authors should cite see Sawaya et al. 2014 PNAS (MR phasing) and Colletier et al. 2016 Nature (de novo phasing).

- line 326. "... requiring much less beamtime for a certain time step" could read "... requiring much less beamtime *and crystalline material* for a certain time step"

- line 333-336. The discussion on "... including "reflections at lower signal-to-noise ratio, for example down to 0.1..." is hand-waving, all the more considering that the authors acknowledge worsening of the data quality when they included reflections with signal-to-noise ratio down to 2 (line 280-282). It should at least be mentioned that serious progress in Laue data processing tools will be required to integrate such signals from polychromatic serial diffraction patterns.

Reviewer #2 (Remarks to the Author):

The manuscript should be published with minor revision.

The following two points still need to be addressed:

"This clearly highlights one of the basic concepts of serial crystallography to improve the achievable resolution as defined by $I/\sigma(I)$ by merging diffraction data from a large number of crystals." According to this logic, one could achieve sub-angstrom resolution if they collected data from an even larger number of crystals. The intrinsic order of the crystals defines the resolution limit. The word "achievable" should be eliminated and the statement should be changed "...to improve completeness at high resolution as ...".

The errors in calibration of individual detector pixels is a cause of error in diffraction measurements, even with PAD detectors. Therefore, the statement "can be regarded as overcome" should be changed to "has been minimized"

Reviewer #3 (Remarks to the Author):

I am writing to you concerning the paper (NCOMMS-17-05571A) that I am refereeing.

The authors replied satisfactorily to my comments and the other reviewers.

This new version of the paper has been improved, and I certainly support the publication of this paper in Nature Communications.

Reviewer #1 (Remarks to the Author):

- line 46. We believe citations should reflect chronological order. Feasibility of structure determination from sub-micron sized-crystals was demonstrated before Gati et al. 2017 PNAS. Authors should cite see Sawaya et al. 2014 PNAS (MR phasing) and Colletier et al. 2016 Nature (de novo phasing).

Reply: Changes were made accordingly

- line 326. "... requiring much less beamtime for a certain time step" could read "... requiring much less beamtime *and crystalline material* for a certain time step"

Reply: Changes were made accordingly

- line 333-336. The discussion on "... including "reflections at lower signal-to-noise ratio, for example down to 0.1..." is hand-waving, all the more considering that the authors acknowledge worsening of the data quality when they included reflections with signal-to-noise ratio down to 2 (line 280-282). It should at least be mentioned that serious progress in Laue data processing tools will be required to integrate such signals from polychromatic serial diffraction patterns.

Reply: Changes were made accordingly

Reviewer #2 (Remarks to the Author):

"This clearly highlights one of the basic concepts of serial crystallography to improve the achievable resolution as defined by $I/\sigma(I)$ by merging diffraction data from a large number of crystals." According to this logic, one could achieve sub-angstrom resolution if they collected data from an even larger number of crystals. The intrinsic order of the crystals defines the resolution limit. The word "achievable" should be eliminated and the statement should be changed "...to improve completeness at high resolution as ...".

Reply: We would like to keep our statement in general here. In order to take the reviewer comment into account we have changes the sentence to: "This clearly highlights one of the basic concepts of serial crystallography where the achievable resolution as defined by $I/\sigma(I)$ can be improved up to the limit of the intrinsic crystal order by merging diffraction data from a large number of crystals."

The errors in calibration of individual detector pixels is a cause of error in diffraction measurements, even with PAD detectors. Therefore, the statement "can be regarded as overcome" should be changed to "has been minimized"

Reply: Changes were made accordingly